# AI-driven automated discovery tools reveal diverse behavioral competencies of biological networks

**Mayalen Etcheverry[1,2], Clément Moulin-Frier[1], Pierre-Yves Oudeyer[1], Michael Levin[3]\***

[1]INRIA, University of Bordeaux, Bordeaux, France; [2]Poietis, Pessac, France; [3]Allen Discovery Center, Tufts University, Medford, United States

## eLife assessment

This **important** study develops a machine learning method to reveal hidden unknown functions and behaviors in gene regulatory networks by searching parameter space in an efficient way. **Solid** evidence is presented for the method, which should be of broad interest to anyone working in biology, as the ideas put forward by the authors extend beyond gene regulatory networks to reveal hidden functions in any complex system with many interacting parts.

**\*For correspondence:**
michael.levin@allencenter.tufts.edu

**Abstract** Many applications in biomedicine and synthetic bioengineering rely on understanding, mapping, predicting, and controlling the complex behavior of chemical and genetic networks. The emerging field of diverse intelligence investigates the problem-solving capacities of unconventional agents. However, few quantitative tools exist for exploring the competencies of non-conventional systems. Here, we view gene regulatory networks (GRNs) as agents navigating a problem space and develop automated tools to map the robust goal states GRNs can reach despite perturbations. Our contributions include: (1) Adapting curiosity-driven exploration algorithms from AI to discover the range of reachable goal states of GRNs, and (2) Proposing empirical tests inspired by behaviorist approaches to assess their navigation competencies. Our data shows that models inferred from biological data can reach a wide spectrum of steady states, exhibiting various competencies in physiological network dynamics without requiring structural changes in network properties or connectivity. We also explore the applicability of these 'behavioral catalogs' for comparing evolved competencies across biological networks, for designing drug interventions in biomedical contexts and synthetic gene networks for bioengineering. These tools and the emphasis on behavior-shaping open new paths for efficiently exploring the complex behavior of biological networks. For the interactive version of this paper, please visit https://developmentalsystems.org/curious-exploration-of-grn-competencies.

## Introduction

Developing methods to recognize, map, predict, and control the complex, context-sensitive behavior of chemical and genetic networks is an essential frontier of research in science and engineering. These systems, such as gene regulatory networks and protein pathways, are known to be instructive drivers of embryogenesis, cell behavior, and complex physiology (*Sanz-Ezquerro et al., 2017*; *Padilla-Longoria Enrique Balleza et al., 2008*; *Huang et al., 2005*). Understanding the control properties of these systems is critical not only for the study of evolutionary developmental biology (*Davidson, 2010*; *Peter and Davidson, 2011*; *Ten Tusscher and Hogeweg, 2011*; *Kim and Sayama, 2018*;

*Srivastava, 2021*), but also for comprehending and intervening in various disease states, including cancer (*Singh et al., 2018*; *Qin et al., 2019*; *Fazilaty et al., 2019*), and for the construction of novel synthetic biologicals in bioengineering contexts (*Davies and Levin, 2022*; *Toda et al., 2018*; *Toda et al., 2020*; *Ho and Morsut, 2021*; *Santorelli et al., 2019*).

Thus, much work has gone into mathematical modeling and computational inference of both protein pathways and gene regulatory network models (*de Jong, 2002*; *Schlitt and Brazma, 2007*; *Fetrow and Babbitt, 2018*; *Delgado and Gómez-Vela, 2019*), which has resulted in the development of large collections of publicly-available models such as the Biomodels database (*Glont, 2018*; *Malik-Sheriff et al., 2020*). Yet, despite the wealth of available models, scientists still largely lack an effective understanding of the range of possible behaviors that these models can exhibit under different initial conditions and environmental stimuli, and are in search of systematic methods to reveal and optimize those behaviors via external interventions. The full extent of the computational and control properties of such networks are not yet well-understood; while dynamical systems theory has been extensively used to characterize their behavior (*Kauffman, 1993*; *Kauffman, 1995*), it is not known what other sets of tools might reveal and exploit interesting properties of this ubiquitous biological substrate. The field of diverse intelligence (also known as basal cognition) has suggested that strong functional symmetries between pathway networks and neural networks could imply the existence of learning and other kinds of behavior in this unconventional substrate (*Abramson and Levin, 2021*; *Baluška and Levin, 2016*; *Dodig-Crnkovic, 2022*; *Dodig-Crnkovic, 2022*; *Timsit and Grégoire, 2021*; *Katz et al., 2018*). Specifically, it has been hypothesized that gene regulatory networks (GRNs) and other molecular networks could be endowed with surprising navigation competencies allowing them to robustly reach diverse homeostatic or allostatic states despite a wide range of perturbations (*Csermely et al., 2020*; *Gyurkó et al., 2013*; *Fields and Levin, 2022*; *Watson et al., 2010*), and that exploiting these innate competencies could provide a promising roadmap for the design of interventions in regenerative medicine and bioengineering contexts (*Mathews et al., 2023*; *Lagasse and Levin, 2023*).

However, significant challenges remain in practice for the exploration and behavior-shaping of these innate competencies, which presents a barrier to the use of these ideas in regenerative medicine and bioengineering. Because of the non-linearity and redundancy in pathway dynamics, passive exploration strategies such as random screening are likely to either fail in uncovering the full range of potential behaviors or require time and energy beyond the available resources. Here, we formalize and investigate a view of gene regulatory networks as agents navigating a problem space. We propose a framework and automated tools, leveraging (1) curiosity-driven goal-directed exploration algorithms coming from recent advances in machine learning and (2) a battery of empirical tests inspired by behaviorist approaches, for mapping the repertoire of robust goal states that GRNs can reach within this problem space despite various perturbations. A key novelty of this work is the use of AI-based exploration tools to map the space of possible behaviors in biological networks, which opens interesting avenues for efficient mapping of unfamiliar system behaviors, yielding transferable insights for diverse problem-solving once such a map is discovered.

The challenge of exploring and mapping spaces of complex and self-organized behaviors appears in many fields such as diverse intelligence in biological systems, minimal active matter, and robotics: many systems in these areas provide a rich space of evolved, engineered, and hybrid systems that offer many of the same fundamental problems of behavior and control regardless of specific composition or provenance (*Clawson and Levin, 2023*). These span many orders of spatio-temporal scale, from molecular assemblies to swarms of complex organisms (*Timsit and Grégoire, 2021*; *Krist et al., 2021*; *Čejková et al., 2017*; *Hanczyc et al., 2011*). One set of approaches seeks to develop tools to identify the optimal level of control, ranging from physical rewiring to various methods from cybernetics and behavioral sciences, to reveal and exploit the native competencies and computational capacities of these systems (*Davies and Levin, 2022*). Specifically, it is increasingly realized that the level of competency (and thus the appropriate level of control) often cannot be guessed by inspection of a system's components, and that its position on a spectrum ranging from passive matter to complex metacognition must be determined empirically (*Rosenblueth et al., 1943*; *Bongard and Levin, 2021Levin, 2022*), (*Clawson and Levin, 2023*). This is critical not only for fundamental understanding of evolution of bodies and minds (*Lyon, 2006*; *Barandiaran and Moreno, 2006*; *Müller and Lengeler, 2000*; *Baluška and Levin, 2016*; *McGivern, 2020*; *Levin, 2023b*), but also for the design of interventions in biomedicine and synthetic morphology contexts (*Pezzulo and Levin, 2015*; *Pezzulo*

*and Levin, 2016*). Yet, a common property in many of these systems is that it is expensive in time and energy to conduct experiments: empirical exploration needs to be made under limited resources. Thus, methods for automating efficient exploration and discovery of a diversity of behaviors in these spaces may be widely useful. As explained below, we will here leverage methods from developmental artificial intelligence initially designed for the specific purpose of exploring a diversity of behaviors using a limited budget of experiments.

One especially fascinating set of systems concerns cellular molecular pathways, or GRNs. In the lab or clinic, these pathways are usually treated as simple machines, with intervention strategies focusing on rewiring their structure to achieve a desired outcome: adding or removing nodes (gene therapy), or changing connection weights (by targeting promoter sequences or protein structures) (*Wong et al., 2008*; *Samuel et al., 2018*; *Krzysztoń et al., 2021*; *Baum, 2007*). However, the emergent, generative nature of development and physiology ensures that it is often very hard to know which genes/proteins to modify, and how, in order to reach a complex desired system-level outcome (*Lobo et al., 2014*). Moreover, the responses of cells and tissues to drugs changes over time, making it even more difficult to infer specific interventions (e.g. drugs) that will induce a stable improvement in pathway state in vivo. Indeed, with the exception of antibiotics and surgery, most available treatment modalities do not solve the underlying problem – they seek to mitigate symptoms, which recur (or expand) once the drug is withdrawn. This is because current therapeutics function bottom-up, attempting to force specific molecular states, as it has been challenging to develop methods for shifting complex tissues and organs towards a stable health profile. Next-generation solutions, which would offer true healing (stable correction), require an understanding of the homeostatic and allostatic properties of networks with respect to how they traverse the space of transcriptional, physiological, and anatomical states. An understanding of the behavior policies of networks as they dynamically navigate these problem spaces is essential for predicting what stimuli can be used to re-set their setpoints and guide them to autonomously maintain a healthy state. In the language of behavioral neuroscience, this strategy corresponds to exploiting their native robustness, decision-making, and navigational competencies to induce predictable, long-lasting changes in functionality.

Significant challenges remain in revealing and controlling the range of behaviors that can self-organize in these cellular and molecular pathways. To characterize steady-state concentrations and responses to small perturbations, conventional methods rely on piecewise-linear approximation of the system behavior (*Stucki, 1979*; *Ingalls, 2004*; *Ingalls, 2008*; *Donzé et al., 2010*; *Dang et al., 2011*), but struggle with higher-dimensional systems or wider parameter ranges which limits their applicability (*Donzé et al., 2011*). Other works have proposed the porting of tools from network control theory to identify sets of control nodes allowing to drive the network behavior toward target steady states (*Rozum and Albert, 2022*). These methods typically exploit the network topology (*Rozum and Albert, 2022*; *Steinway et al., 2015*; *Zañudo and Albert, 2015*; *Zañudo et al., 2017*; *Fontanals et al., 2020*) or regulatory structure (*Murrugarra et al., 2016*; *Choo et al., 2018*; *Choo et al., 2019*) to identify control strategies based either on permanent knockout/activation of genes or on temporary perturbations, the latter being preferable in biomedical context.

However, these approaches often require prior knowledge of target attractor states or are limited to Boolean network models. Other works have explored the use of machine learning tools, such as evolutionary search (*Paladugu et al., 2006*; *François, 2014*; *Noman et al., 2015*) and gradient-descent optimization (*Hiscock, 2019*; *Shen et al., 2021*), for controlling continuous ODE biomolecular networks with high-dimensional parameter spaces, mainly in the context of synthetic circuit engineering (*Camacho et al., 2018*; *Volk et al., 2020*). While providing powerful optimization tools, these approaches tend to focus on rewiring network structure and connectivity. Moreover, the choice of a predefined fitness function and parameter range initialization is not only critical to the success of optimization (*François, 2014*) but largely restricts the exploration of the behavior space (*Shen et al., 2021*).

In contrast, an alternative line of research proposes exploring and leveraging the inherent molecular mechanisms of adaptivity and robustness in cellular pathways as a promising approach for drug interventions that do not rely on genomic editing or gene therapy (*Csermely et al., 2020*; *Kitano, 2007a*). Recently, a broad, substrate-independent behavior science perspective suggests novel properties of GRNs and other biological networks (*Abramson and Levin, 2021*; *Manicka and Levin, 2019*). This perspective views GRNs as agents that convert activation levels of specific genes (inputs)

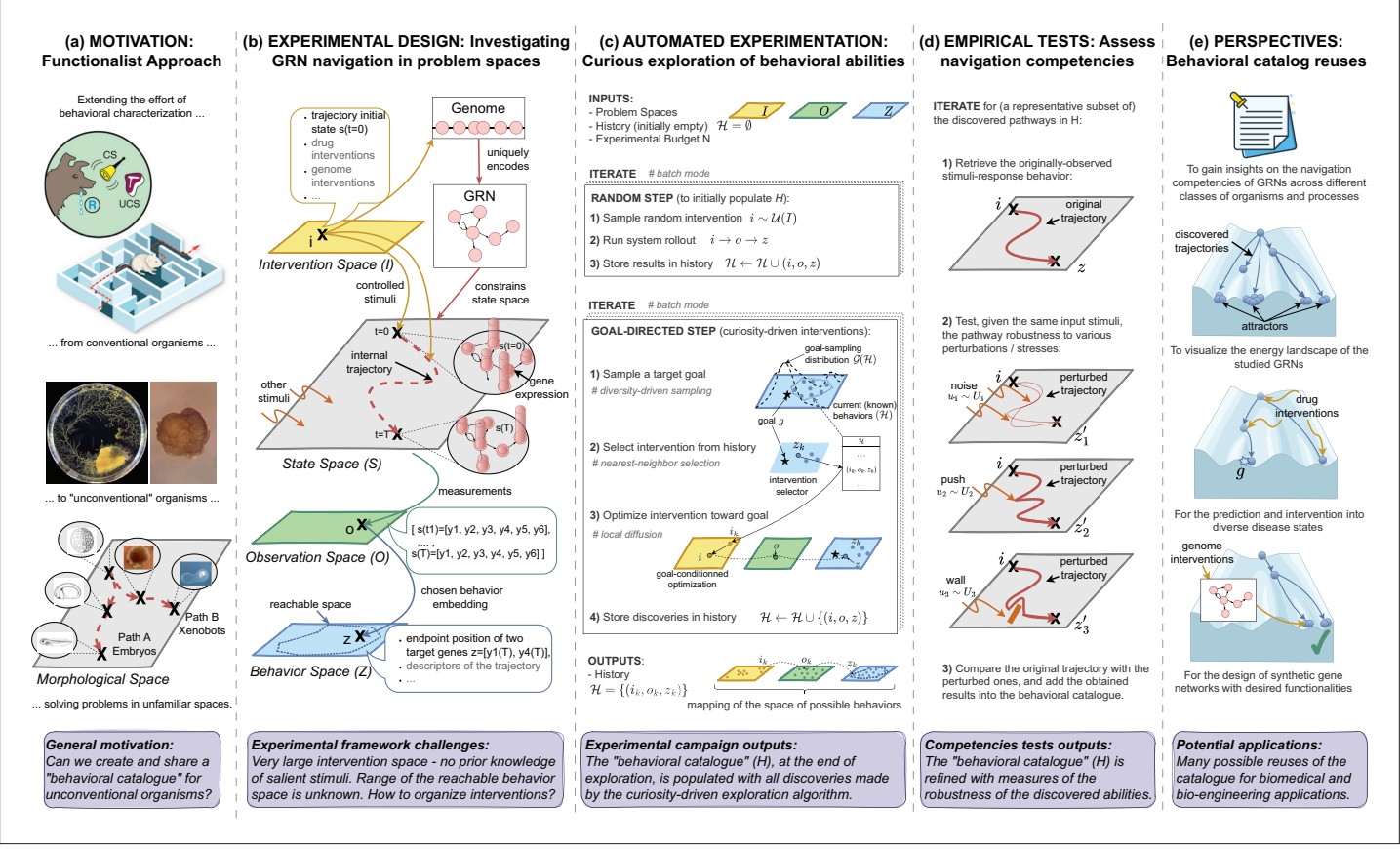

**Figure 1.** Overview of the proposed framework. (**a**) MOTIVATION: We often focus on studying the navigation and behavior of organisms in conventional three-dimensional environments, neglecting the intelligence underlying competencies at sub-organismal scales (*Fields and Levin, 2022*). To better understand navigation competencies in unconventional organisms solving problems in unconventional spaces (e.g., embryos in morphological space), it is essential to construct comprehensive 'behavioral catalogs' for these novel entities, which in turn requires sophisticated exploration methods to discover the extent of possible behaviors. Images are taken and adapted with permissions from Figure 2C in *Levin, 2022*, Figure 6A in *Levin, 2023a* (image made by Alexis Pietak), graphical abstract of *Murugan et al., 2021*, Figure 3D of *Kriegman et al., 2020* (image made by Douglas Blackiston) and Figure 4A of *Bongard and Levin, 2023*. (**b**) EXPERIMENTAL DESIGNS: We formalize gene regulatory network (GRN) behavior as a navigation task and propose to investigate it by defining abstract and observer-dependent 'problem spaces' that we use to organize the observed biological behaviors and their exploration in practice. (**c**) AUTOMATED EXPERIMENTATION: Pseudo-code of the curiosity-driven goal exploration process we use to automate the discovery of behavioral abilities that the GRN can exhibit in behavior space. (**d**) EMPIRICAL TESTS: We use a battery of empirical tests to identify the robust goal states of the systems, i.e., the one that can be attained under a wide variety of perturbation (including noise in gene expression, and pushes or walls during traversal of transcription space). (**e**) PERSPECTIVES: We explore several potential reuses of the discovered 'behavioral catalog' and proposed framework across evolutionary biology, biomedicine, and bioengineering contexts.

to those of effector genes (outputs), with intermediate nodes in between, leading to strategies for controlling network behavior based on a specific history of inputs (experience) rather than through network rewiring. Notably, the concept of training a chemical pathway using pulsed input stimuli (node activation or suppression drugs) has been formalized, and several networks have been analyzed to establish a taxonomy of memory types found in biological GRNs and pathways (*Biswas et al., 2021*; *Biswas et al., 2023*).

Here, building upon recent research (*Fields and Levin, 2022*; *Biswas et al., 2021*; *Biswas et al., 2023*), we take the next step and investigate a view of gene regulatory networks as agents navigating a problem space toward target goal states with varying degrees of competency (*Figure 1a*). We seek to implement a definition of goal that abstracts it from conventional associations with human or other advanced brains and facilitates the use of tools from cybernetics, behavior science, and control theory to understand broader aspects of biological regulation. Here, we use the term 'goal' state to refer to

a system's steady state, which it expends effort to reach despite interventions or barriers - a definition appropriate to the study of basal (or minimal) proto-cognitive regulatory systems. Our definition of goal does not imply 'purpose' (high-level goals where an agent has the meta-cognition to think about having goals and what they might be), and we do not attribute high-level competencies (such as re-setting one's own goals) to GRNs.

Our particular focus lies in investigating two types of navigation competencies: *versatility*, which refers to the capacity to reach diverse goal states under different interventions, and *robustness*, which refers to the ability to reach a goal state despite various perturbations. The primary scientific question we aim to address is: What is the repertoire of robust goal states that a GRN can actively reach through *minimal* and *non-genetic* interventions within a navigation task context, and can we develop systematic methods and automated tools to aid scientists in discovering this repertoire?

To address this question in practice, our experimental framework revolves around the definition of 'problem spaces,' which we use as tractable components of the GRN's overall state space (*Figure 1b*), and on a set of methodological contributions which we organize around three sub-questions:

1. *Automated discovery of diverse behavioral abilities with autotelic curiosity search* (*Figure 1c*): What is the range of possible goal states that GRNs can exhibit and how can we devise efficient exploration strategies to automatically identify these goal states? Defining goal states as attractor states of the underlying gene regulatory network, we show that traditional screening methods can be very inefficient in discovering the range of possible goal states. To address this, we propose to use intrinsically-motivated goal exploration processes (IMGEP) (*Baranes and Oudeyer, 2013*; *Forestier et al., 2022*), a recent family of diversity-driven machine learning approaches also known as autotelic curiosity search which was recently shown to form a useful discovery assistant for revealing the behavioral diversity of unfamiliar systems such as chemical oil-droplet systems (*Grizou et al., 2020*), physical non-equilibrium systems (*Falk et al., 2024*) and models of continuous cellular automata (*Reinke et al., 2020*; *Etcheverry et al., 2020*; *Hamon et al., 2024*).

2. *Evaluation of the navigation competencies* (*Figure 1d*): How competent is the GRN, in terms of robustness to perturbations, in attaining the diverse previously-identified goal states? Prior studies have offered definitions of robustness in biological networks, characterized as the degree of variation in functionality (*Kitano, 2007b*) or phenotypic trait (*Félix and Barkoulas, 2015*) under specific environmental or genetic changes. However, these studies often consider a predefined functionality and random perturbations in network parameters (*Ingolia, 2004*; *Ma et al., 2006*; *Noman et al., 2015*) or specific gene knockouts (*Deutscher et al., 2006*). Environmental perturbations on the other hand are often limited to random variations in initial conditions within a predefined range (*Donzé et al., 2011*; *von Dassow et al., 2000*). Here, inspired by behaviorist approaches, we test hypotheses about non-genetic resistance with respect to various navigation competencies that living agents often exhibit, and that do not require structural changes of network properties or connectivity. Those tests assess the system's ability to maintain robustness despite various perturbations encountered during traversal, including developmental noise in gene expression levels, sudden 'pushes' within transcriptional space, and the presence of energy barriers or 'walls' acting as force fields in the environment.

3. *Potential reuses of the discovered 'behavioral catalog' and framework* (*Figure 1e*): Can the constructed behavioral catalogs be useful for fundamental research and practical therapeutic applications, and can the framework be easily applied to other systems and problem spaces? We propose that the discovered competencies may provide valuable insights for understanding evolvability and developmental robustness, and provide a fertile source for the design of interventions in biomedicine and synthetic morphology contexts. We also suggest that the framework and automated tools, which are observer-focused and substrate-independent, could be transposed to other systems and problem spaces.

The overall framework is summarized in *Figure 1*. Applying it on a database of 30 continuous (ODE) models from the Biomodels website, consisting of a total of 432 systems defined as GRN model-behavior space tuples, revealed several interesting insights. First, results suggested that most of the surveyed systems are capable of reaching a surprisingly wide spectrum of steady states depending on their initial state. Interestingly, random screening strategies were not able to reveal this diversity of reachable states (or at least not in a sample efficient way), confirming the need for more advanced exploration strategies like curiosity search. Second, among the discovered steady states, we were able to identify several robust goal states i.e., ones that the system consistently reaches despite various

perturbations during traversal of transcriptional space. Altogether, these findings seem to suggest that cell phenotype and functionality could be the result of a multi-step program (*Steinway et al., 2015*) that could be flexibly and robustly reprogrammed by appropriate stimuli (*Levin, 2022*). Finally, we demonstrate possible reuses of this 'behavioral catalog' for comparing the network's competencies across different classes of organisms, as well as for the design of non-genetic drug interventions. We also demonstrate an alternative reuse of the framework to reveal new kinds of reachable 'goals' in synthetic gene networks, suggesting alternative strategies for the design of gene networks in a bioengineering context.

An interactive executable version of the paper, as well as step-by-step tutorials and notebooks, can be found online at https://developmentalsystems.org/curious-exploration-of-grn-competencies. The full codebase of the proposed automated experimentation pipeline is written end-to-end in JAX, a high-performance numerical computing library that we leverage for parallel experimentation and computational speedups of the ODE models time-course simulations.

## Results

### Generalizing GRN behavior as a navigation task

The GRNs analyzed in this study are biological pathway networks taken from the BioModels repository (*Glont, 2018*; *Malik-Sheriff et al., 2020*). The term 'GRN' is used broadly to include protein interaction, gene regulatory, and metabolic networks. In these mathematical models, the dynamic interactions between nodes of the network (molecular species) are modeled with a system of ordinary differential equations, enabling to quantitatively simulate time-course behavior (model rollouts) and observe the dynamics of node activities over time (*Figure 2a*). Here, following a terminology which aims to integrate concepts from dynamical complex systems with concepts from behavioral sciences, we propose to conceptualize GRN behavior as a *navigation task* (*Table 1*). Model rollouts are viewed as 'trajectories' in transcriptional space where network steady states are 'goal states' (endpoints) that the 'agent' (GRN) can reach with varying levels of competencies. As for living agents, these competencies may range from unstable locomotion patterns to more advanced forms of goal-directed behavior like path following, obstacle avoidance, or even forms of spatial memory and foresight. In this paper, we are particularly interested in investigating two forms of navigation competencies that we refer to as *versatility*, the capacity to reach diverse goal states under various interventions, and *robustness*, the capacity to reach a goal state despite various perturbations. Note that versatility and robustness are studied with respect to different sources of incoming environmental variation, respectively interventions and perturbations.

To investigate these competencies in practice, our experimental framework is based on the definition of 'problem spaces,' which include the observation space (O), behavior space (Z), intervention space (I), and perturbation space (U) as defined in *Table 2*. To be consistent with our navigation task terminology introduced in *Table 1*, we refer to a behavior z as the reached 'goal state' of a GRN trajectory. However these 'goals' may lie on a continuum between complete robustness and high sensitivity, and our primary interest lies in identifying *robust* goals of the system. Whereas several choices could be made for the intervention space I and perturbation space U, we intentionally consider *minimal* and *non-genetic* interventions to investigate the 'native' goal states of the GRN, and *environmental obstacles* to investigate for navigation competencies classically observed in other living agents. Examples of simulations, interventions, and perturbations are illustrated in *Figure 2*.

Then, a typical analysis using our framework relies on a 2-step procedure, detailed in the subsequent sections. First, to assess the versatility of the GRN, we define an exploration strategy which organizes the sequence of interventions $i_1, \ldots, i_N$ used to drive the system toward a maximally diverse set of reachable endpoints $\{z_k \in Z\}_{k=1..N}$, while being given a limited budget of experiments N. Second, to assess the robustness of the discovered goal states $\{z_k \in Z\}$, we conduct a battery of empirical tests to characterize their degree of sensitivity to novel perturbations, with a fixed experimental budget of P perturbations per selected behavior z. At the end of this two-step procedure, we obtain the 'behavioral catalog' (H) of the studied GRN, which includes the history of experiments $H = \left\{ \left( i_k, o_k, z_k, \left\{ \left( u_p, o_p, z_p \right), p = 1...P \right\} \right), k = 1 \ldots N \right\}$.

Following this framework, the behavioral catalog is constructed for a database of 30 biological networks consisting of a total of 432 systems, where a system is defined as a (GRN model, intervention

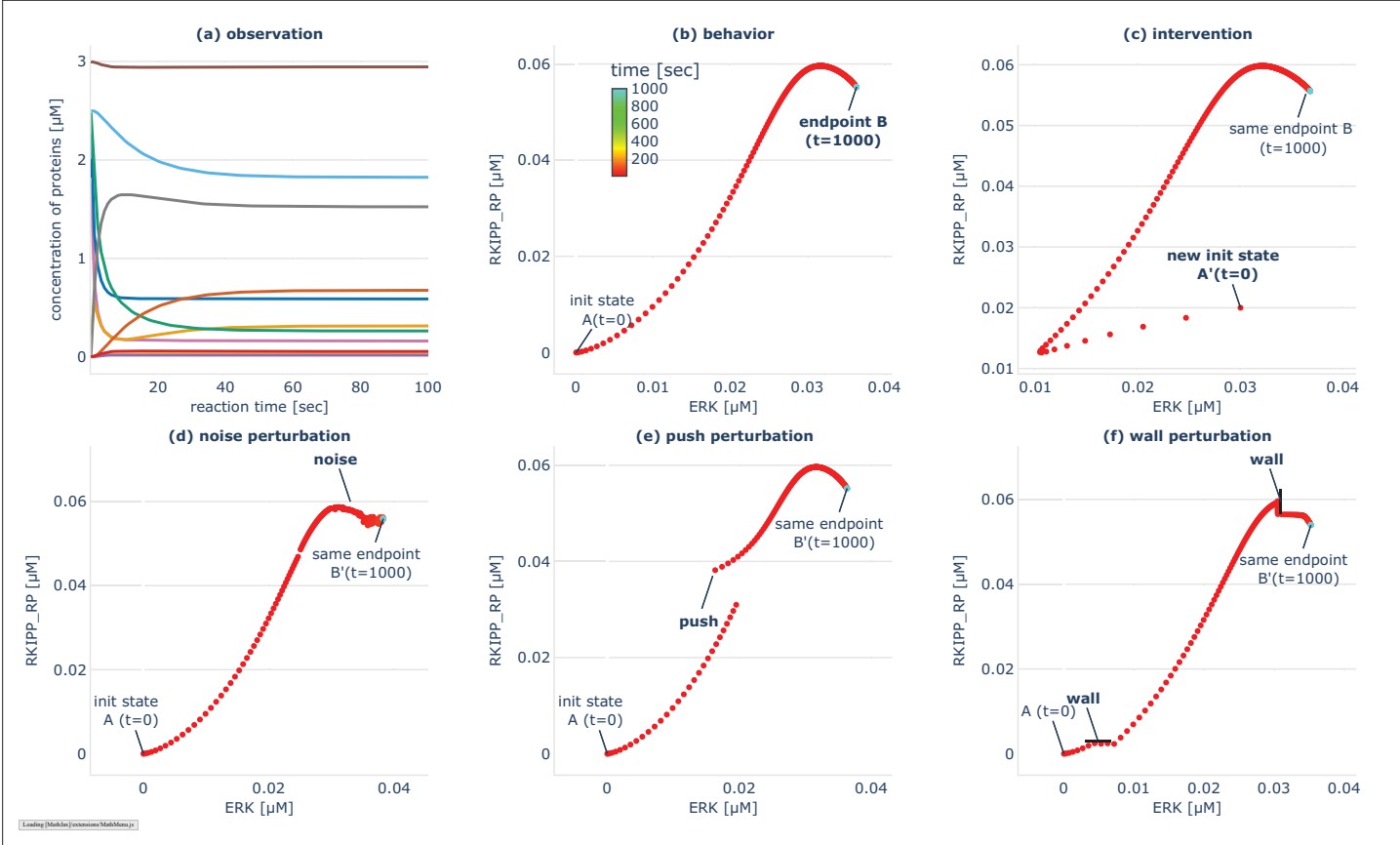

**Figure 2.** Illustration of the experimental setup and chosen problem spaces on an example gene regulatory network (GRN) model which has 10 nodes and models the influence of RKIP on the ERK Signaling Pathway (**Lehman and Stanley, 2011**). (**a**) Time-course evolution of the different nodes y1, …, y10 (one color per node) when starting from the default initial conditions (as provided in **Lehman and Stanley, 2011**). The observation captures the states taken through time o=[y(t=0), …, y(t=T)] where y=[y1, …, y10]. (**b**) Corresponding trajectory in transcriptional space (phase space), for two target nodes (ERK, RKIPP_RP), from t=0 (A, in red) to T=1000 s (B, in cyan). We can see that the trajectory converges to endpoint B in less than 100 s, and then stay there. The behavior (or reached goal state) is the endpoint $B = \left[ y_{ERK}\left(T\right), y_{RKIPRP}\left(T\right) \right]$, where T is chosen big enough to ensure convergence. (**c**) The intervention is setting the initial state of the system trajectory (for all nodes): i = [y1(t=0), …, y10(t=0)]. (**d-e**) Example of perturbations used in this paper. (**d**) Noise perturbation, here applied to all 10 nodes every 5 s until t=80 s. (**e**) Push perturbation, here applied to the two target nodes (ERK, RKIPP_RP) at t=3 s. (**f**) Wall perturbation, also applied to the two target nodes (ERK, RKIPP_RP), here at 10% and 90% of the total distance traveled. Figure supplement shows examples of other possible 'drug' or 'genome' interventions that can be implemented in the accompanying software, as well as the possibility to perform interventions (or perturbations) in parallel using batched computations.

The online version of this article includes the following figure supplement(s) for figure 2:

**Figure supplement 1.** Examples of interventions that can be implemented within the accompanying AutodiscJax software.

space (I), behavior space (Z)) tuple, as described in Materials and Methods and **Supplementary file 1**. These catalogs provide valuable empirical observations and insights into the navigation competencies of the studied GRNs, particularly in their ability to consistently achieve diverse goal states under various tested perturbations. Statistical analyses of the results are presented in **Figure 3**; Figure 5; Figure 7, and specific results for the RKIP-ERK signaling pathway (**Kwang-Hyun et al., 2003**) are shown in **Figure 2**; **Figure 4**, Figures 6, 8.

## Curiosity search uncovers a diversity of reachable goal states

One advantage of modeling GRN behavior within a tractable behavior space Z is that we can then deploy strategies to efficiently discover and map that space. Notably, recent diversity-driven machine learning techniques such as Novelty Search (**Lehman and Stanley, 2008**; **Lehman and Stanley, 2011**), Quality Diversity (**Cully et al., 2015**; **Pugh et al., 2016**) and Intrinsically-Motivated Goal Exploration Processes (IMGEP) (**Baranes and Oudeyer, 2013**; **Forestier et al., 2022**) are explicitly designed to efficiently explore a so-called 'behavior space' or 'goal space' which is basically a (predefined or

**Table 1.** Glossary of terms used in this paper, with the proposed isomorphism which investigates concepts from dynamical complex systems and behavioral sciences under a common navigation task perspective.

| Dynamical systems terminology | Behavioral science terminology | Proposed isomorphism | Navigation task terminology |
|---|---|---|---|
| **system**: a set of interconnected elements that interact to produce emergent behavior | **organism**: a living being that responds to stimuli and adapts to its environment | Both are collections of lower-level elements that interact to produce emergent behavior and can adapt at the system level | **agent** or GRN |
| **phase-space trajectory**: set of states taken by the system when starting from one particular initial condition | **behavioral trajectory**: the sequence of states that an organism exhibits in response to stimuli | Both represent the sequence of states or behaviors that a system or individual experiences over time | **trajectory** |
| **initial condition**: initial state of a system's variables and parameters that condition its dynamics | **stimuli**: events that might (or might not) trigger a response in an organism | Both represent incoming variations that set a system or organism in motion | **intervention** or **perturbation** |
| **critical parameter**: a parameter or condition that, if changed, can cause a system to undergo a qualitative change or phase transition | **salient stimuli:** stimuli that are particularly relevant or meaningful to an organism, either because they are associated with reward or punishment or because they are novel or unexpected | Both represent the incoming variations that have a significant impact on a system's steady-state or organism's response | **salient intervention** |
| **steady-state (or attractor)**: a stable state (or set of states), towards which the system tends to evolve over time | **observed response**: outcome or endpoint of a behavioral trajectory towards which an organism converges | Both represent the endpoint that a system or organism is moving towards | **reached endpoint** or **goal** |
| **robust attractor**: stable attractor toward which the system tends to evolve under various initial conditions and perturbations | **target goal**: it is assumed that an organism engages in a goal-directed manner when it exhibits new ways or actions to achieve a similar outcome when faced with novel circumstances | Both represent a stable endpoint or goal that the system successfully attains under various perturbations | **robust goal** |
| **controllability**: degree to which the system's dynamics (and resulting steady states) can be controlled or manipulated | **trainability:** degree to which an organism's behavior can be modified or shaped by experience or conditioning | Both measure the extent of states that can be reached by a system or individual under a distribution of stimuli/conditions | **versatility** |

learned) model of the overall state space. In particular IMGEPs, which were originally developed for the learning of inverse models of highly-redundant mapping in robotics context (*Baranes and Oudeyer, 2013*), were recently shown to successfully assist discovery in complex self-organizing systems (*Grizou et al., 2020*; *Falk et al., 2024*; *Reinke et al., 2020*; *Etcheverry et al., 2020*).

Here, we propose to use an IMGEP to control GRN initial states and maximize the diversity of discovered endpoints $\{z \in Z\}$ within a limited budget of $N$ experiments. The IMGEP operates in two phases: initially, $N_f$ interventions are sampled randomly from $I$ to populate history $H$, then remaining interventions are generated through a goal-directed process which relies on several key internal models. Those including a goal-embedding module $(R)$ that encodes observations $(o)$ into the IMGEP goal space (), a goal generator module $(G)$ that samples goals from the goal space based on intrinsic motivation incentives (e.g. to promote novelty or learning progress), and a goal-conditioned optimization policy $(\Pi)$ that generates candidate intervention parameters to achieve the current goal. Given those internal models, the goal-directed phase iterates through (1) sample a target goal $g \sim G(H)$, (2) infer intervention parameters to achieve that goal $i \sim \Pi(g, H)$, (3) conduct an experiment with the intervention i, observe the outcome o, and compute the reached goal $z = R(o)$, and (4) store the tuple

**Table 2.** Problem spaces used in this study.

| Problem space | Generic definition | Specific definition in this study |
|---|---|---|
| Observation space (O) | Space of raw observations made during the GRN model rollout to measure its state or behavior. | Records node activities over time as $o = \big(y(0), \ldots, y(T)\big)$, where y(t) is an n-dimensional vector (n=number of nodes) and T is the measured reaction time. |
| Behavior space (Z) | A projection of the observation space used by the experimenter to encode the 'goal states' of a model rollout into a tractable (lower-dimensional) space. | Encodes the trajectory endpoint of a model rollout. Represents a cell phenotype defined by the state values of some nodes (relevant biological markers), such that $z = \big(y_{i1}(T), \cdots y_I(T)\big)$ (we use m=2 in this study for simplicity and visualization). |
| Intervention space (I) | A space where interventions represent controlled sources of incoming variation that the experimenter can exert on the GRN model rollout to drive it toward novel or targeted states. | Sets the initial state $i = \big(y_1(0), \ldots, y_n(0)\big)$ of a model rollout. Defined as a hyper-rectangle I ⊆ Rⁿ where the boundaries are proportional to the min and max values taken by the respective nodes from default initial conditions. |
| Perturbation space (U) | A space where perturbations represent external sources of incoming variation, used by the experimenter to characterize the robustness of a given goal state. | Includes three classes of (stochastic) perturbations including noise perturbation $U_n$, push perturbation $U_p$, and wall perturbation $U_w$. |

$(i, o, z)$ in history $H$. Here, we use the GRN behavior space Z as the IMGEP goal space $= Z$. Hence 'target goal' refers to a goal sampled by IMGEP while 'reached goal' refers to an actual endpoint of the GRN trajectory, discovered by IMGEP while targeting a potentially different point in Z. Throughout exploration, the IMGEP dynamically refines its Z-traversal strategy based on the knowledge acquired by its discoveries. Here, we opt for a simple IMGEP variant such that the exploration process can be seen as performing novelty search in behavior space Z (**Doncieux et al., 2019**). The pseudocode of our IMGEP pipeline is shown in **Figure 1c** and details about the internal models are provided in Materials and Methods. The final outcome is a 'behavioral catalog' of the GRN, containing the diverse goal states discovered by IMGEP: $H = \big\{(i_k, o_k, z_k), k = 1 \ldots N\big\}$.

We deploy the IMGEP, also known as 'curiosity search,' on all 432 systems in the biological network database. Our evaluation focuses on two related competencies: the IMGEP agent's ability to empirically reveal a diversity of reachable goal states in the (GRN, I, Z) system, referred to as 'discovered diversity,' and the GRN agent's competency to naturally reach diverse goal states, referred to as 'versatility.' The true versatility of the GRN is unknown and can only be inferred through empirical exploration and proxy metrics.

For evaluating diversity, we measure the area covered in Z by the IMGEP discoveries using the threshold-coverage metric (**Benureau, 2015**) and compare it with the area covered by the diversity of a naive random screening strategy (which uniformly samples interventions in $I$). In **Figure 3**, the diversity discovered by the two exploration variants is shown for the 432 $(GRN, I, Z)$ systems, where random search is given a budget of experiments $N$ which is twice bigger (n=900) as the one given to the curiosity-search algorithm (n=450). Interestingly we see that, on average, at n=290 the curiosity search already significantly outperforms the final diversity achieved by random search, while only utilizing one-third of its experimental budget (n=900). Whereas we are dealing with numerical systems and our codebase allow for efficient and parallel execution, each experiment still consists of $\frac{T}{\Delta T} = 25000$ model steps, where each step integrates the ODE system. Repeating that N times for each of the 432 systems starts to be very costly, which is why having efficient exploration strategies is very valuable (and would be even more valuable when scaling the framework to larger databases). Even more critical, as illustrated in **Figure 3b**, it seems that, for some systems, random search is not able to discover the 'latent' regions revealed by the IMGEP in Z, or it would need an extremely large budget of experiments. On the other hand, as illustrated in **Figure 3c**, there are some systems for which random search is already quite efficient in revealing diverse behaviors in Z, and for which IMGEP performs equivalently.

In fact, the goal-directed strategy of the IMGEP is particularly beneficial for $(GRN, I, Z)$ systems with high nonlinearity or redundancy in their $I \rightarrow Z$ mapping, as seen in **Figure 4** and studied in robotics contexts (**Benureau, 2015**). Redundancy implies that many interventions in $I$ lead to similar effects in $Z$, as illustrated in **Figure 2** where various interventions and perturbations converge to the same endpoint. In these systems, random search will preferentially discover points in areas of high

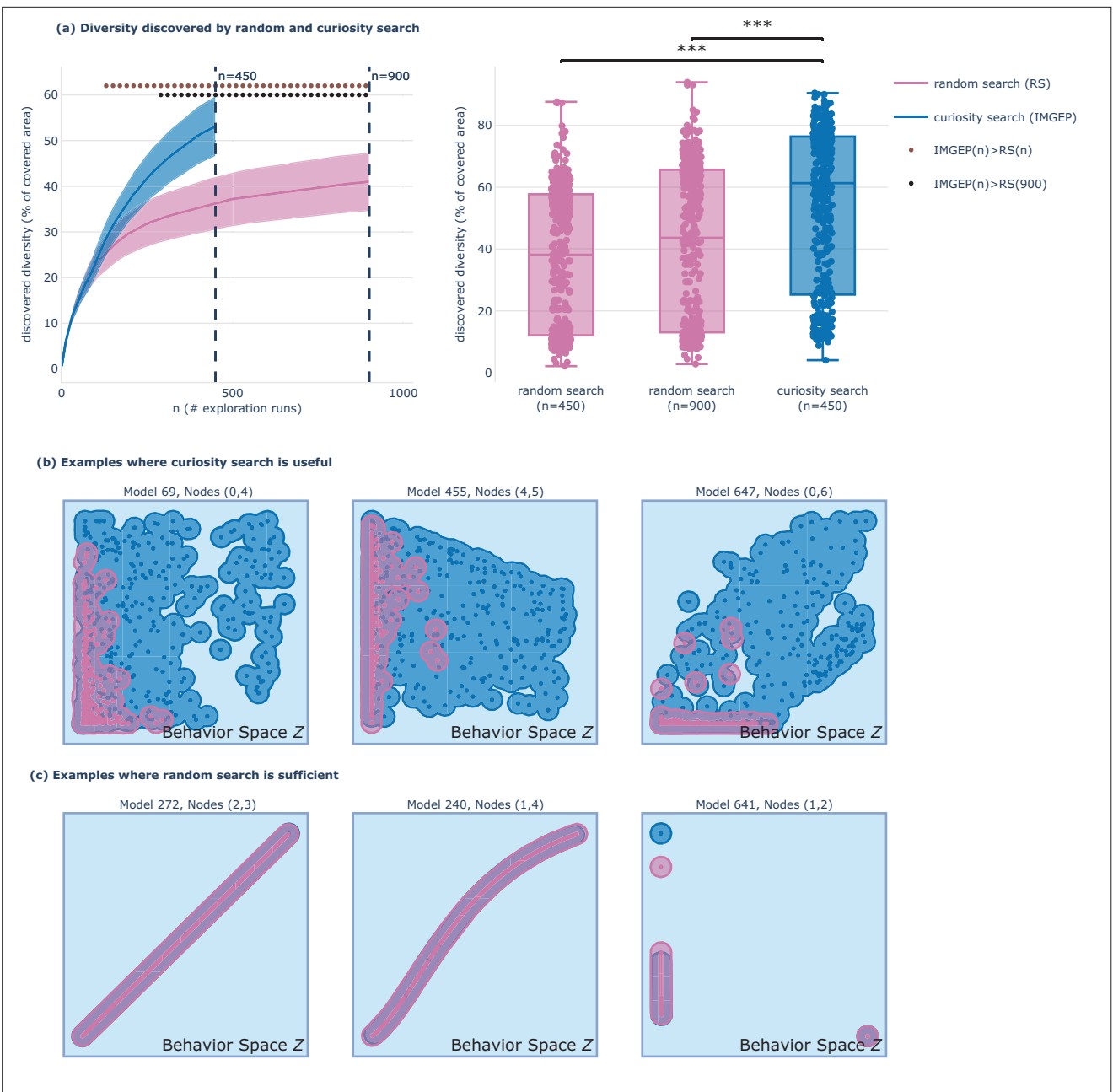

**Figure 3.** Curiosity search uncovers a wide spectrum of reachable states in behavior space Z. (**a**) Diversity of endpoints discovered by random search (pink) and curiosity search (blue) for the 432 systems. Diversity is measured as the volume of the union of the set of hyperballs of radius $\epsilon$ that have for centers the discovered endpoints $\{z \in Z\}$ as depicted by the shaded area in (**b–c**) with $\epsilon = 0.05$. (a-left) Mean and standard deviation curves of the diversity of behaviors discovered throughout exploration (with random search having twice more experiments n=900). Dots indicates significance (p<0.05) when testing curiosity search (**n**) against random search (**n**) in brown, and against random search (n=900) in black, with a Welch's t-test. Standard deviation is divided by 4 for visibility. (a-right) Detail of the diversity obtained in the left plot for all 432 systems at n=450 and n=900, where *** indicates significance (p<0.001). (**b–c**) Discovered endpoints at the end of exploration (n=450) by random search (pink) and curiosity search (blue) for 6 example systems of our database. (**b**) Examples of systems for which curiosity search is much more sample-efficient than random search in finding a diversity of reachable states in behavior space Z. (**c**) Examples of systems with low-redundancy mapping I ->Z such that random search in $I$ is already quite efficient in covering behavior space Z, and curiosity search performs equivalently.

The online version of this article includes the following figure supplement(s) for figure 3:

**Figure supplement 1.** Sanity Check.

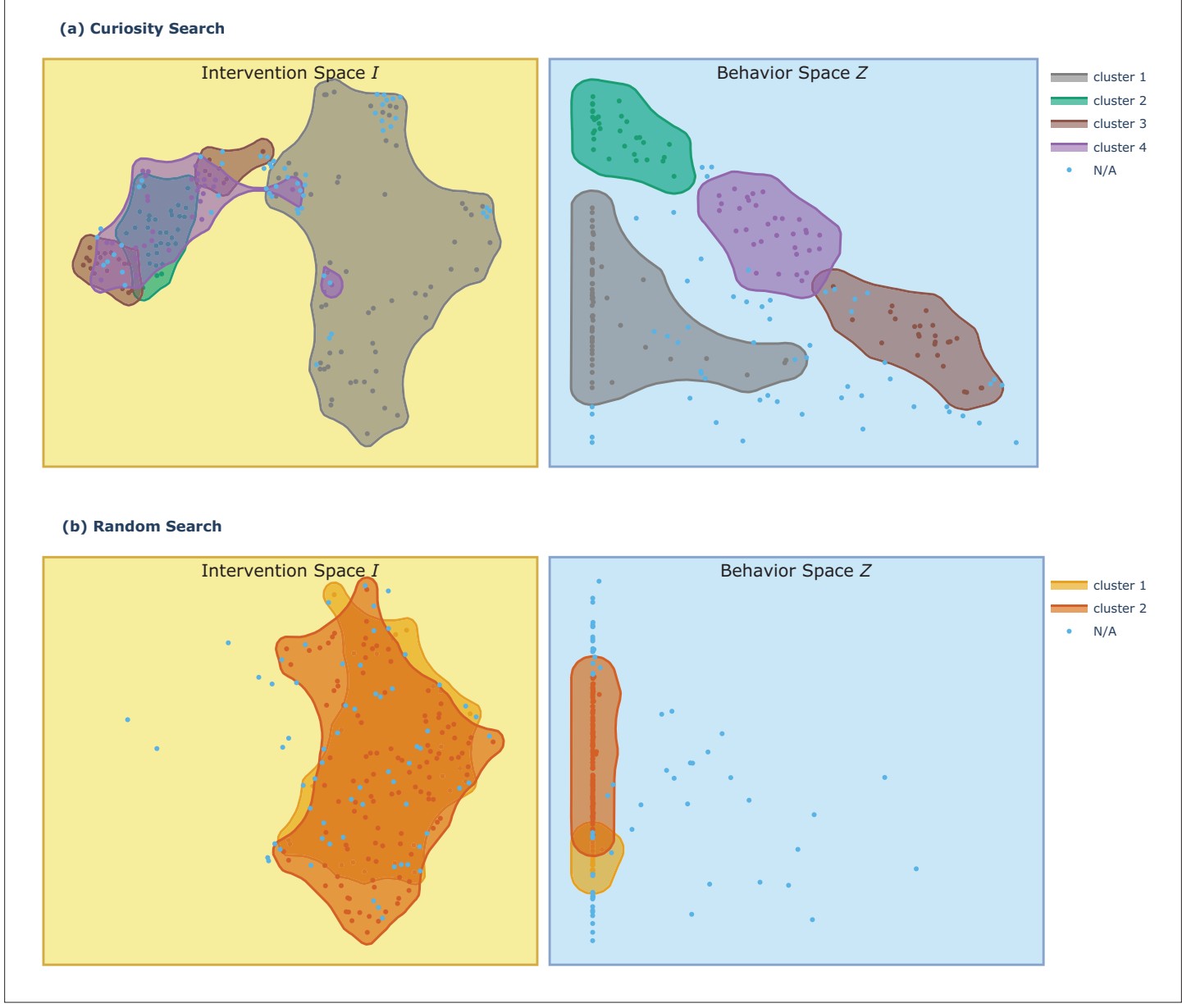

**Figure 4.** Illustration of the non linearity and redundancy of the I->Z mapping, and of the interest of using goal-directed exploration strategies. Plot shows the reachable points discovered by curiosity search (**a**) and by random search (**b**) in the behavior space Z and their corresponding starting points in the intervention space I, for the RKIP-ERK signaling pathway system (***Kwang-Hyun et al., 2003***). The intervention space is 10-dimensional, and here we show the TSNE reduction in 2D. We apply HDBSCAN clustering (***McInnes et al., 2017***) on the points discovered in Z, which produced four clusters for curiosity search (displayed in gray, green, purple, and orange; non-assigned points are displayed in light blue) and two clusters for random search (displayed in light and dark orange). We then visualize where those regions in behavior space are mapped back in the intervention space, by applying the same coloring. (**a**) Looking at the curiosity search discoveries, we can see the non-linearity of the I->Z mapping, where small regions of intervention space can map to large regions of the behavior space (like the orange area) and reversely (gray area). We can also see the redundancy of the behavior space which is clearly concentrated in the left border of the space (ERK close to zero) which can seemingly be reached from very large portions of the intervention space (gray area). (**b**) Looking at random search discoveries, we can understand that it is very inefficient as it spends most of its exploration budget in the region of intervention space that converges to the left border in Z, and fails to explore the orange, purple, and green regions discovered by curiosity search which seemingly lead to the more novelty in Z.

redundancy in Z whereas the IMGEP, whose exploration is directed uniformly in goal space, should cover different levels of redundancy equally. In general, when dealing with large intervention spaces and limited experimental budgets, curiosity search can be particularly useful for efficiently navigating Z-space.

Finally, as the IMGEP efficiently drives the GRN into diverse goal states with minimal interventions, we propose that the diversity achieved by the IMGEP can serve as a good proxy metric of the GRN versatility. Notably, analysis of example systems in *Figure 3* reveals that many GRNs can reach a broad spectrum of steady states. Whereas our database is limited to certain systems (see Materials and methods) and might not be representative of all biological pathways, this observation underlines the existence of various phenotypes that can be realized. It also highlights the critical importance of identifying salient interventions that can effectively control cellular states within this spectrum of possibilities, notably as many cancer types are due to epigenetically non-identical cells (*Bell and Gilan, 2020*).

## Empirical tests reveal robust navigation competencies

We are then interested in characterizing the degree of robustness of the previously-discovered 'goal states' in order to identify the ones that can consistently be reached by the GRN despite encountering various perturbations. Whereas many studies have proposed rigorous analysis of the 'robustness' of biological networks (*Kitano, 2007b*; *Félix and Barkoulas, 2015*), the generated perturbations often target variations in the regulatory rules (i.e. variations at the hardware level) and variations are often sampled independently (and prior) to observations of the GRN dynamical behaviors (*von Dassow et al., 2000*; *Ingolia, 2004*; *Ma et al., 2006*; *Noman et al., 2015*; *Rizk et al., 2009*; *Donzé et al., 2011*). Here instead, we propose to conduct a battery of empirical tests that draw inspiration from classical 'displacement experiments' (*Walcott, 1996*; *Luschi et al., 2001*) and 'barrier experiments' (*Bisch-Knaden and Wehner, 2001*) commonly used in behavioral sciences to assess the navigation competencies of various animals. As illustrated in *Figure 2*, we consider *environmental* perturbations that perturb the GRN trajectory with (1) various degree of noise in the gene expression levels, (2) sudden 'pushes' during the GRN traversal of transcriptional space, and (3) energy barriers or 'walls' acting as new force fields that constrain the GRN traversal. Importantly, those perturbations are *conditioned* on the observed behavior of the GRN. The magnitude of the noise and of the pushes is scaled proportionally to the extent of the observed trajectories, and the walls are generated in locations of the space that the GRN would 'naturally' visit without the induced perturbation. While intuitive from a behaviorist point of view, where one would adapt experimentation when testing animals in different contexts (e.g. to study homing behavior of an ant and of a sea turtle, or of an ant in food deprivation and in reproduction phase) (*Abramson, 1994*), robustness studies in systems biology often neglect those aspects. We propose that a behaviorist lens on robustness can help in understanding forms of non-genetic resistance in transcriptional space, which is crucial for the development of therapeutic strategies (*Bell and Gilan, 2020*).

To assess the degree of robustness of the discovered goal states, our evaluation procedure is the following. For each (GRN, I, Z) system of the database, we retrieve a representative set of trajectories previously discovered using the curiosity-search algorithm and subject these trajectories to $P = s \times r$ perturbations conditioned on the GRN goal-reaching trajectory $i \rightarrow z$ prior perturbation. Here, s represents the different perturbation distributions which correspond to various 'tests' and 'levels of difficulty' (e.g. noise magnitude and frequency, number of walls, etc.) and $r$ is the number of (stochastic) perturbations sampled per family. The pseudocode is illustrated in *Figure 1c* and details about the different family of perturbations are provided in Materials and Methods. At the end of this process, the behavioral catalog is augmented with the perturbed trajectories $H = \left\{ \left( i_k, o_k, z_k, \left\{ \left( u_p, o_p, z_p \right), p = 1...P \right\} \right), k = 1 \dots K \right\}$.

As the use of 'spaces' comes with the notion of similarity and distance, we can then easily evaluate the *sensitivity* of a goal state $z$ with respect to a set of perturbation $\left\{ u_p, p = 1...P \right\}$ as the average distance in behavior space Z between the original trajectory endpoint $z$ and the perturbed trajectories endpoints $\left\{ z_p \right\}$. Here our distance is simply the Euclidean distance, normalized by the extent of the trajectory prior perturbation in Z. We can then identify the so-called 'robust goals' of the systems as the ones that have the lower sensitivity to perturbations. These sensitivity analyses can be useful in two important ways. On the one hand, they allow us to quickly identify the 'extreme' examples of robustness, both at the system-level and at the goal-level, providing several insights into the degree of 'competencies' that some biological networks might exhibit in their relative space (*Figure 5*). On the other hand, these analyses also allow us to map the *heterogeneity* of cellular responses and to better understand how non-genetic perturbations might modulate the *landscape* of reachable cell phenotypes (*Figure 6*).

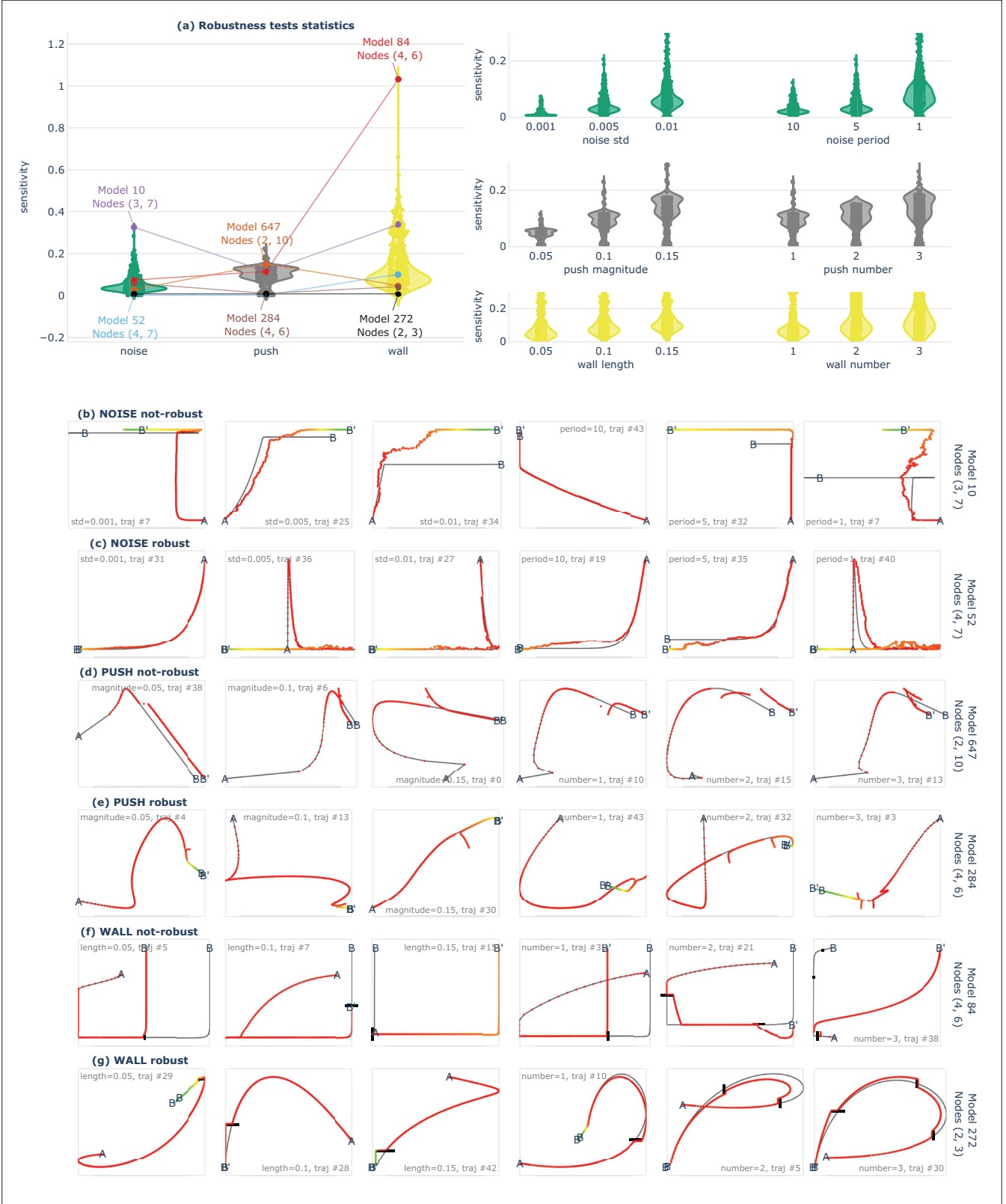

**Figure 5.** Identification of robust traversal strategies in transcriptional space. (**a**) Violin plots show, for each of the 432 systems (one point per system), the median sensitivity (over the K representative goal states) to the noise (green), push (gray), and wall (yellow) perturbation families. Violin plots on the right detail the median sensitivity for the 18 sub-families. (**b–g**) Each row provides examples of perturbed trajectories of either extremely-robust or extremely-sensitive example (gene regulatory network. GRN, Z) system (on average over the K goal states) for the three families of perturbations, as

*Figure 5 continued on next page*

*Figure 5 continued*

shown by annotations in (**a**). For instance, the first row (**b**) shows perturbed trajectories of the (model #10, nodes (3, 7)) system which has the highest sensitivity to noise whereas the last row (**g**) shows trajectories of the (model #272, nodes (2, 3)) system which has a nearly perfect robustness to walls. Each image contains an example trajectory for a given $(i, u)$, and one $u$ per sub-family is shown per column. For instance, in the first row (**b**), the trajectories are perturbed with the different sub-families of noise ($\sigma_n \in [0.001, 0.005, 0.1]$, $p_n \in [10, 5, 1]$) which can be seen as various levels of difficulty. For each trajectory we annotate the starting position (A), endpoint prior perturbation (B), and endpoint after perturbation (B'), and show the original trajectory in black. The perturbed trajectory is shown in color scale (from red at t=0 to cyan at t=3000 s). (**b**) Except for few cases (trajectory #43), the system (model #10, nodes (3, 7)) system is not robust to noise as its trajectories are easily deviated from the original endpoint. (**c**) The (model #52, nodes (4, 7)) system however, except for rare cases (trajectory #35), consistently reaches its original target despite encountering various amounts of noise. Interestingly, trajectories #36 and #40 consistently follows a complex up->right-down->left path, despite the induced noise. (**d**) The (model #647, nodes (2, 10)) system, except for few cases (trajectory #0), is typically deviated from its original trajectory when being pushed away. Interestingly though, it seems to follow similar (parallel) trajectories. (**e**) The (model #284, nodes (4, 6)) system, is an example of an extremely robust system which, despite many push configurations (in magnitude and number), consistently returns to its original trajectory. Interestingly, the trajectories of this system are relatively complex with several loops and detours. (**f**) The (model #84, nodes (4, 6)) system is not very robust to walls, and typically deviates or blocked when it encounters a wall. (**g**) The (model #272, nodes (2, 3)) system is another example of an extremely robust system which, despite many wall configurations (in length and number), consistently returns to its original path. Once again interestingly, the trajectories of this system are relatively complex with several loops and detours.

The online version of this article includes the following figure supplement(s) for figure 5:

**Figure supplement 1.** Wall implementation.

---

*Figure 5* shows the median sensitivity, over the representative goal states, for the 432 systems of our database and for the noise, push, and wall perturbations families (as well as for the s=18 sub-families which correspond to varying degrees of perturbations). Overall, even though we observe varying degrees of sensitivity between systems (and between magnitudes of perturbations, which is expected), one first and interesting observation is that the median sensitivity remains relatively low, suggesting that GRNs could not only exhibit versatility (with respect to the considered interventions) but also robustness (with respect to the considered perturbations). In fact, looking at the 'extreme' examples, we can identify quite impressive examples of complex and yet highly-robust space traversal strategies, with non-linear trajectories exhibiting many 'detours' and 'loops' but yet consistently reaching the same endpoint despite several pushes (*Figure 5e*) or walls (*Figure 5g*) on the way.

---

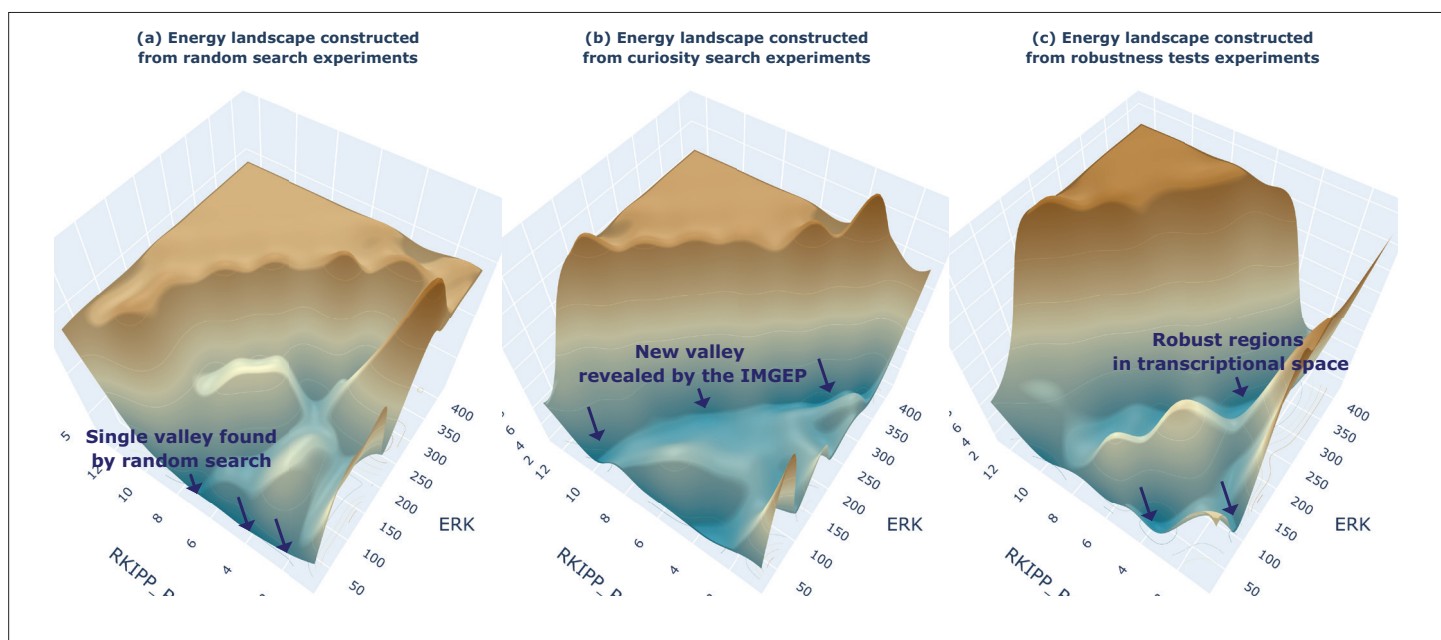

**Figure 6.** Visualization of the system-level energy landscape. Energy landscape visualization based on the trajectory-based landscape generation method (*Li and Wang, 2013*), and constructed from different sets of gene regulatory network (GRN) trajectories, respectively trajectories generated (**a**) by the random search exploration, (**b**) by the curiosity-driven exploration, and (**c**) by the robustness tests experiments.

*Figure 6* shows how the constructed catalog $H$ can be used to generate the energy landscape of the studied system. In biology, landscape formalisms have been used to comprehend the underlying dynamics of several systems, such as cell cycles and cell differentiation (*Li and Wang, 2014*; *Li and Wang, 2013*). It is believed that such system-level visualizations could be particularly useful to apprehend non-genetic heterogeneity in the context of cancer treatment and stem cell differentiation (*Bell and Gilan, 2020*; *Venkatachalapathy et al., 2021*). A recent landscape-generation method only proposes to approximate the pseudopotential energy through simulation trajectories obtained throughout exploration of the system (*Venkatachalapathy et al., 2021*), making it a widely applicable method which we can directly apply here. However, the paper relied on Monte Carlo simulation to generate the trajectories. Due to the previously mentioned non-linearity and redundancy of the I->Z mapping, this can lead to poor estimation of the overall energy landscape (*Figure 6a*). Instead, when generating the landscape from the trajectories discovered by our curiosity search exploration, we are able to reveal a new and wide 'valley' of reachable states (*Figure 6b*). Interestingly, the landscape-generation method can also be used to better comprehend the effect of external cues on the gene regulatory network, by visualizing how much they deform the energy landscape for instance leading to new shaped valleys (*Figure 6c*). For the example system RKIP-ERK pathway (*Kwang-Hyun et al., 2003*), results highlighted a specific region of behavior space (with low RKIP and high ERK activation levels) that seems to be particularly robust, i.e., consistently reached by the GRN from certain initial conditions, and that might be associated with tumor development (*Lee et al., 2006*).

## Possible reuses of the behavioral catalog and framework

Our framework generated a catalog of stimuli, responses, and navigation tests for the different GRN models contained in our database. Creating and sharing such a 'behavioral catalog' with the scientific community is possibly one of the more exciting aspects of the work with new organisms. Furnished with such an empirically based data-set and detailed observations, one can (1) conduct statistical analysis across the population of studied organisms to inform fundamental research questions and (2) reuse

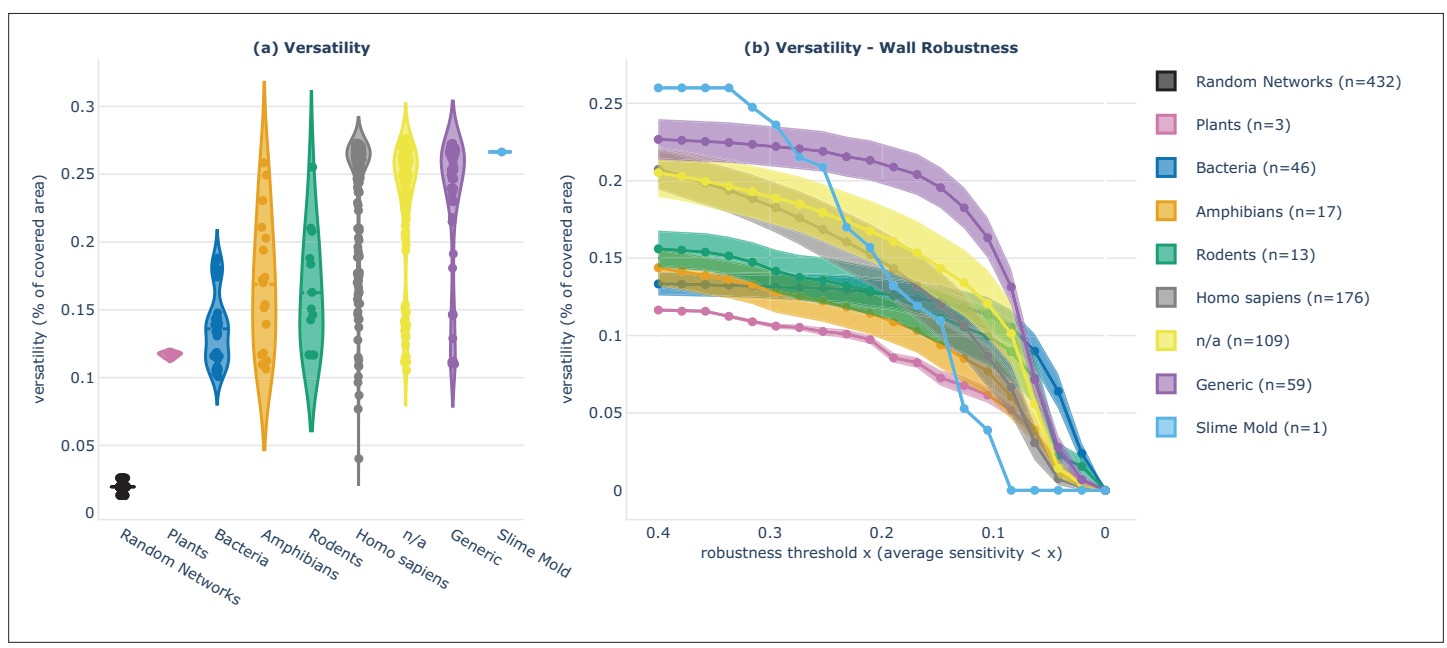

**Figure 7.** Analysis and comparison of the degree of sophistication, in terms of versatility and robustness, between different classes of gene regulatory network (GRN). We categorize the GRNs by class of organism they belong to: plant, bacteria, slime mold, amphibian, rodent, *Homo sapiens*, or generic. 'n/a' refers to network models for which this information is not available. (**a**) Violin plots show the versatility of the 432 systems (one point per system) for each class. Versatility of one system is measured as the area covered by all the goal states discovered by curiosity search (equivalent to what we call diversity in *Figure 3*). (**b**) Trade-off (aka Pareto) mean and standard deviation curves that represent the trade-off among versatility and wall robustness performances as taken by the different classes of GRNs (standard deviation is divided by 4 for visibility). For each system, versatility (y-value) is measured as the area covered by the set of robustly achieved goal states, where the criterion of goal-achievement is a binary which tests whether the goal-reaching sensitivity (on average overall wall perturbations) is below a certain threshold (x-values). Violin plots in (**a**) are ordered in ascending order according to the class mean y-value at x=0.4 in (**b**).

the acquired knowledge to design specific behavior-shaping experiments in organisms of interest. As our framework focuses on observable behavior and is agnostic about the internal construction of the organism, another exciting perspective is to deploy it to different problem spaces and other classes of natural, chimeric, or synthetic organisms. This section illustrates preliminary experiments along those three types of reuse.

## To develop insights on the degree of sophistication of the different GRNs

The first use-case we explore is to conduct statistical analysis to categorize versatility and robustness in the surveyed networks on the basis of species in evolutionary strata. We consider seven categories, namely, plant, bacteria, slime mold, amphibian, rodent, *Homo sapiens*, or generic. Here, generic corresponds to the networks not associated with any species but related to generalized biological processes. Please note that the surveyed database is relatively small with respect to the wealth of available models and biological pathways, so we can hardly claim that these results represent the true distribution of competencies across these organism categories. Still, as shown in *Figure 7*, results suggested interesting patterns.

First, on average, generic and *Homo sapiens* GRNs exhibit higher versatility (mean 0.228 and 0.238) compared to rodent and amphibian GRNs (mean 0.163 and 0.169), which in turn show higher versatility than bacteria and plant GRNs (mean 0.136 and 0.117). These findings are particularly intriguing in the context of the recently-formulated hypothesis of multi-scale competency architecture (*Levin, 2022*): could the observed variation in versatility among different classes of GRNs contribute to the degree of versatility observed at higher-level scales? Collecting such experimental data for broader classes of organisms and GRNs will be crucial to understand how competencies at the molecular scale can impact the overall functionality and adaptability of organisms at higher scales, and to understand how evolution might have exploited this modular architecture for shaping the observed adaptivity and reprogrammability of biological systems.

Second, when comparing with the versatility of random networks (in black), generated to follow the same distributions of network size and connectivity as biological networks (as proposed in *Biswas et al., 2023*, see Materials and Methods), we observe that random network versatility is much lower (<0.026) than the versatility observed in biological networks. Once again, it is difficult to draw strong conclusions as the gene circuit model used for the random networks is relatively limited, whilst generic and studied across a range of biological contexts (*Reinitz and Sharp, 1995*; *Jaeger et al., 2004*; *Cotterell and Sharpe, 2010*; *Molinelli et al., 2013*), and it will be interesting to scale the comparison to a broader and more complex range of ODE-based random models. Still, these findings hint that versatility prevalence might be a strong invariant of biological intelligence shaped by evolutionary processes.

Finally, we categorize the versatility-robustness tradeoff in the different categories of organisms. The idea is to compare the GRN competencies to robustly achieve diverse goal states, for different robustness thresholds. In *Figure 7b*, we plot the mean and standard deviation pareto curves for the different categories of organisms and observe that, in average, the pareto-optimal solutions are mostly achieved by generic cell GRNs, even though bacteria GRNs can robustly reach more goal states for exigent robustness criteria (high x-values). The slime mold GRN can reach highly diverse goal states but the tradeoff quickly drops with wall perturbations, and there is only one system in our database belonging to this category so results might be not representative. Once again, those results are very interesting as generic cells GRNs are a building block that has been extensively reused by evolution across several organisms and contexts, bacteria have evolved to be very resistant (e.g. to antibiotics), and slime molds are an unicellular organism well known for its diverse capabilities, especially navigational ones (*Vallverdú et al., 2018*; *Beekman and Latty, 2015*; *Saigusa et al., 2008*; *Nakagaki and Guy, 2008*).

## For the development of therapeutic interventions

Understanding forms of non-genetic resistance and non-genetic heterogeneity is crucial across a wide range of cancer and treatment contexts (*Bell and Gilan, 2020*). Here, we illustrate how the constructed behavioral catalog can provide a fertile source for the design of therapeutic strategies, notably in the context of network control, using again the example of the RKIP-ERK signaling

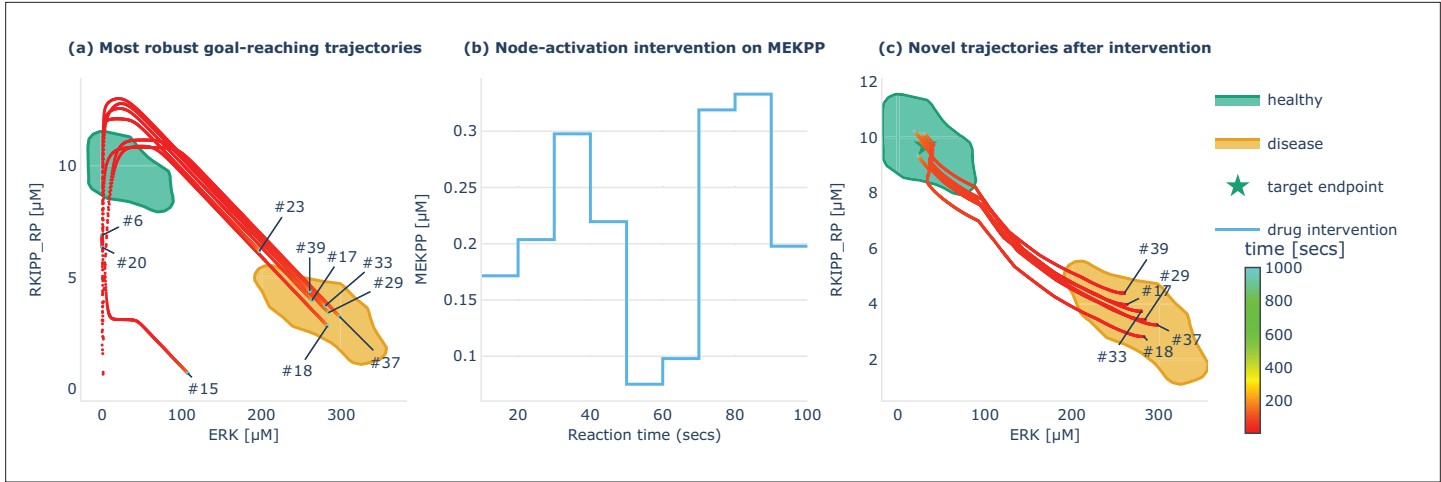

**Figure 8.** Identification of stimuli-based stepwise intervention triggering robust re-set of disease states into healthy physiological states. (**a**) The 10 most robust identified goal states (average sensitivity <0.05) and the corresponding reaching trajectories are displayed for the example RKIP-ERK signaling pathway (*Kwang-Hyun et al., 2003*). We can see that most of them converge toward attractors in the 'disease' region (orange). (**b**) Discovered stepwise stimuli intervention on MEKPP which we apply on states stuck in the 'disease' region for 100 s. (**c**) The discovered intervention successfully brings back all points from the 'disease' region closer to the target endpoint in the 'healthy' region, and this is under various tested perturbations (as shown in figure supplement). The optimization procedure that led to the discovery of this intervention is described in the main text.

The online version of this article includes the following figure supplement(s) for figure 8:

**Figure supplement 1.** Resulting trajectories after applying the discovered stimuli-based intervention, as shown in *Figure 8b*, to the example RKIP-ERK signaling pathway (*Kwang-Hyun et al., 2003*) for the 6 'disease' trajectories originally discovered in the behavioral catalog (shown in *Figure 8a*).

pathway (*Kwang-Hyun et al., 2003*). In *Figure 4*, we saw that curiosity search revealed four clusters of reachable steady states for this system. From a clinical perspective, one might denote the green cluster as 'healthy' region of behavior space and the orange cluster as 'disease' region of the behavior space, as high levels of ERK and low-levels of RKIP are often linked to tumor development (*Lee et al., 2006*). In *Figure 8a*, we plot those two clusters as well as the 10 more robust goal-reaching behaviors from the behavioral catalog of this system, i.e., the goal states with the lower average sensitivity to the induced perturbations. We see that 6 out of the 10 more robust trajectories end up in the 'disease' region, suggesting that certain configurations of initial state are very likely to reach that region despite chemical blockers (here pushes, walls, and noise), which was also visible on the system's energy landscape in *Figure 6c*. Looking at the six trajectories, it seems that they all follow similar patterns where RKIP activation level increases past a certain threshold, and only then converge toward the disease region. This might already provide an interesting biomarker for prediction of tumor development, but what we are really interested here is to build upon that knowledge to develop stimuli-based interventions allowing to re-set the gene regulatory network (GRN) setpoints from the identified 'disease' steady states back to steady states within the identified 'healthy' region. To do so, we define a parameterized stimuli-based intervention and a performance function, and search for parameters that optimize this performance. For the intervention function, we use a piecewise constant function that determines which nodes to intervene on (here MEKPP), when to apply the intervention (here every 10 s for 100 s), and with what amplitude (which are the parameters that we are seeking to optimize). The choice of the intervention function, which is arbitrary in this example, would typically depend on the experimental constraints, e.g., which nodes can be targeted with drugs and at which precision. For the performance function, we define the centroid of the 'healthy' region as the target setpoint and compute performance of the stepwise intervention as the average distance of the novel setpoints (after intervention when starting from the 6 disease setpoints) to the target setpoint, and under a distribution of stochastic walls, pushes and noise perturbations. Hence a successful intervention should re-set the disease setpoints to healthy setpoints for all discovered disease states and robustly across the various tested perturbations. For optimization, we simply perform random search as this was sufficient here to discover one intervention (as shown in *Figure 8b*) that successfully reset the setpoints (as shown in *Figure 8c*) under various tested perturbations (as

shown in *Figure 8—figure supplement 1*). Here random search was sufficient to find a successful intervention, but more advanced optimization strategies like evolutionary algorithms or stochastic gradient descent could be envisaged for harder problems. Overall, mapping the 'latent' behavioral abilities of GRNs in healthy physiology and disease states may have important implications for the identification of robust stimuli-based interventions that focus on behavior shaping instead of micro-managing all molecular states, and that can be exploited in therapeutic contexts.

## As an alternative strategy to gene circuit engineering

The final type of reuse we explore is not a direct reuse of the constructed behavioral catalogs, but rather a reuse of the proposed automated tools to reveal different kinds of behaviors in a bioengineering context. A common problem in synthetic biology is to optimize the configuration and parameters of a gene model network to optimally perform a desired functionality, also known as gene circuit engineering (*Volk et al., 2020*). Recent approaches rely on optimization-driven machine-learning strategies, such as evolutionary algorithms and stochastic gradient descent. However, choosing the right loss function and parameter initialization for these optimization methods is a well-known problem in machine learning. These issues can lead to optimization algorithms getting trapped in local minima within the complex landscape of possibilities. In response to these challenges, we propose to investigate whether the curiosity-driven exploration strategy can be employed as an alternative (diversity-driven) strategy. Whereas traditionally-employed for exploratory purposes, these exploration strategies were also shown to facilitate the resolution of external, pre-defined tasks characterized by sparse or deceptive rewards (*Colas et al., 2018*), by effectively exploring solution space.

Here, we consider the target application of oscillator circuit engineering followed in *Hiscock, 2019*, where parameters of a gene circuit model are optimized to produce oscillation patterns with target amplitude $A$, frequency $w$ and offset $b$. This time, the intervention space includes both genetic interventions (setting kinematic parameters of regulatory rules) and environmental interventions (setting the initial state $y_0$). We then compare several exploration strategies: a random search, two optimization-driven strategies, one using gradient descent as proposed in *Hiscock, 2019* and one using an evolutionary algorithm, and finally a diversity-driven strategy using the proposed curiosity search. All algorithms are given the same experimental budget $(N = 5000)$. For curiosity search, the behavior space $Z$ is defined as the image space of the discrete Fourier transform of the observation. We then use the exact same IMGEP algorithm as before, but operating within the novel problem spaces $(I, Z)$. For gradient descent, we follow the procedure proposed in *Hiscock, 2019*. We define a loss function which measures the mean square error between the observed node activation levels $y$ and the target oscillation (represented as a cosine wave). We then randomly initialize the parameters $i \sim U(I)$ and use Adam optimizer for n=5000 optimization steps. For the evolutionary algorithm, we use the CMA-ES algorithm (*Hansen et al., 2003*), with the negative of the loss as fitness function. In addition, we also use gradient descent for local refinement of the best discoveries made by the other exploration strategies (curiosity search and random search), this time with a limited budget of $N = 100$ optimization steps.

In *Figure 9*, we show that curiosity search is again significantly more efficient than random search in revealing a diversity of possible oscillator behaviors. Out of 5000 trials, random search was able to find only 42 configurations leading to sustained oscillations whereas curiosity search was able to find 1167 (and gradient descent did not find any). Without focusing on the target objective, curiosity search is able to efficiently cover the analytic $(A, \omega, b)$ space (*Figure 9a-c*), thus discovering oscillators close to the target one (*Figure 9d*). Instead, when starting from a random initial condition, gradient descent is very likely to get trapped in a local minimum where it converges to the target offset $b$ but fails to produce oscillations (*Figure 9h-f*). While CMA-ES explores more the solution space at the beginning of optimization than SGD does, it also ultimately converges into a similar local minima. However, whereas the global optimization strategies are unsuccessful in this example, they seem to be useful to locally refine close-enough solutions, as can be seen here when refining the best discoveries made by curiosity search and random search with gradient descent (*Figure 9i-j*). These results suggest that a diversity-driven exploration strategy, eventually combined with a more advanced local optimization strategy, can offer promising and cost-effective alternatives for the design of synthetic gene networks. More generally, as our framework only relies on empirical investigation for inferring the mapping between interventions and behaviors (treating them as abstract variables in observable

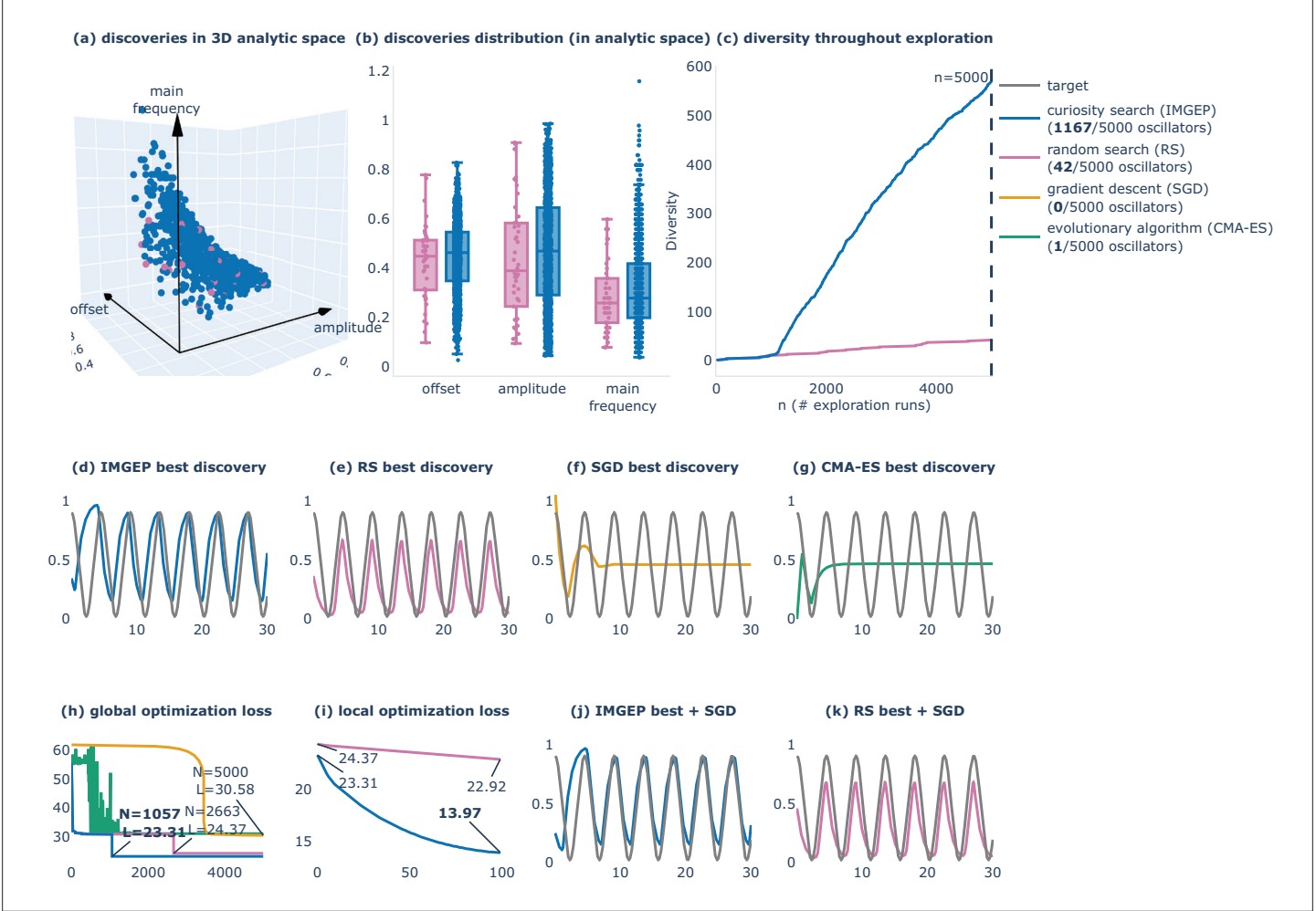

**Figure 9.** Comparison of four alternative strategies for the design of oscillator circuits: curiosity search (blue), random search (pink), gradient descent (orange), and an evolutionary algorithm (green). (a–c) Given a budget of 5000 experiments, curiosity search is able to find 1167 oscillator circuits (ones showing sustained oscillations), whereas random search only finds 42 oscillators and optimization-driven search fail to discover them (only one discovered by CMA-ES and none for gradient descent when starting from a single random initialization). (a) 3D scatter plot of the 42 random search discoveries (pink) and 1167 curiosity search ones (blue) in the (amplitude, main frequency, offset) analytic behavior space. (b) Box plots projecting points from the 3D scatter plot into the respective (amplitude, main frequency, offset) axes. (c) Diversity is discovered throughout exploration, where diversity is measured with a binning-based space coverage metric (20 bins per dimension). (d-e-f-g) Best discoveries (for which is minimal) made by the four exploration strategies. (h) Evolution of the optimization loss $L$ for the four algorithm variants. (i) Evolution of the local training loss when finetuning of the best Intrinsically-Motivated Goal Exploration Processes (IMGEP) (blue) and RS (pink) discoveries with gradient descent, with the finetuned results displayed in (j-k).

problem spaces), we believe it offers an exciting perspective to be deployed across various problem spaces and classes of organisms.

## Discussion

This paper presents a novel framework aimed at uncovering the navigation competencies of GRNs. The framework conceptualizes GRNs as agents actively navigating the transcriptional space and provides a set of tools, leveraging computational models of curiosity-driven learning and exploration, with a battery of empirical tests inspired from behaviorist tradition, for automated experimentation and behavioral characterization. The proposed framework is novel in two central ways. First, it introduces a novel AI-based toolbox to the field of biological network analysis. We show how this toolbox, leveraging the successful ingredients of recent intrinsically motivated learning algorithms - originally developed to enable robotic AI agents to explore and learn diverse skills in novel and unstructured

environments (*Baranes and Oudeyer, 2013*; *Forestier et al., 2022*) - can be transposed to assist efficient discovery of behavioral abilities within biological pathway models like GRNs. Second, rather than merely mapping the attractor states (*Kauffman, 1993*; *Kauffman, 1995*; *Dang et al., 2011*) or analyzing their sensitivity to model parameter changes (*Ingalls, 2004*; *Ingalls, 2008*) as extensively proposed in conventional GRN analysis methods, our framework investigates the dynamic adaptability of these networks' navigation competencies in response to various changing environmental conditions. With this approach, our aim is to uncover whether diverse competencies, analogous to the ones exhibited by living agents, can be found within physiological network dynamics. Notably, these competencies are discovered without necessitating structural alterations to network properties or connectivity. Importantly, our framework and its associated tools do not make any assumptions about the structure or origin of the biological network, making it in theory adaptable to the study of diverse unconventional intelligences across various domains.

By applying this framework to a curated database of GRN models, we discovered a diverse range of behavioral responses that GRN can exhibit under different initial conditions and characterized their robustness to various perturbations. Notably, our analysis revealed a number of interesting aspects of navigation of the state space which can be leveraged in several contexts. These automated tools form the first step towards cost-effective in silico simulation and interrogation platforms; as the 'behavioral catalogs' produced by this process can be a first stepping stone for better understanding the GRN functionalities as well as for designing drug-driven interventions in a biomedical or bioengineering context.

There are several limitations and avenues for future work on this study. First, these networks are studied as a model in isolation and it is possible that some of the ODE models (or solvers) provide spurious behaviors within certain parameter ranges which might not map to observable phenotypes in vitro. Interestingly, this limitation also suggests an interesting further direction to this work: using the automated discovery toolbox to assist model inference, allowing to efficiently identify the rare or unexpected behaviors of the ODE model and suggest whether further refinement is needed or not. Another interesting direction for future work, as our framework considers the GRN model as a black-box and works with limited experimental budget, would be to directly apply it to in vitro GRN models at the bench. One could for instance integrate experimental constraints to the search by defining families of empirically-testable interventions and perturbations, as well as specify clinically-relevant goal spaces and perturbations. Even if in a biological setting versatility and robustness phenomena may be harder to detect, or harder to alter, these results can be used to (1) design synthetic biology circuits with advanced capabilities (*Pandi et al., 2022*), and (2) conduct studies of subcellular proto-cognitive phylogenetics, to help understand the evolutionary pressures for and against reprogrammability in cell regulatory machinery. Another limitation of our work is that we consider predefined problem spaces, here the space of GRN steady states (or Fourier descriptors of the dynamics in the bioengineering example). The dynamics of gene regulatory networks are relatively simple (usually converge to stable points or periodic orbits) allowing such hand-defined descriptors. To scale the framework to higher-dimensional and more complex problem spaces, recent works from the IMGEP literature suggest using unsupervised learning of goal space representations (*Reinke et al., 2020*; *Etcheverry et al., 2020*). Whereas these works were applied to abstract models of multicellular patterning, similar works could be envisaged in more realistic systems, such as sophisticated model of multicellular morphogen and/or bioelectrical patterning which were used to suggest in-vitro experimental manipulations (*Libby et al., 2019*; *Pietak and Levin, 2016*; *Pietak and Levin, 2016*).

The tools presented here, and the behavioral repertoire we identified, are just the beginning, and much work remains. Future efforts must test additional competencies across the spectrum of cognition (memory, creative problem-solving, valence, etc.) and extend the tools we presented here to explore them. The predictions made by our computational tools can now be tested in real cells, using emerging tools for physiological profiling in the living state and a diverse set of biochemical, biomechanical, and bioelectrical perturbations. We anticipate a tight and productive feedback loop between computational theory that suggests new experiments, and results in living cells that greatly extend our computational perspective on what they can do (*Koseska and Bastiaens, 2017*; *Baluška et al., 2022*; *Baluška et al., 2023*; *Reber and Baluška, 2021*; *Baluška and Reber, 2021*). Such interdisciplinary work, pulling together concepts and techniques across fields, is likely to have major implications for fundamental understanding of evolution, intelligence, and dynamical control, as well as

drive novel kinds of therapeutics that leverage the innate behavioral competencies of living matter (*Levin, 2023b*; *Bernheim-Groswasser et al., 2018*).

## Methods

### GRN models and numerical simulation

This study employs ordinary differential equation (ODE) models to represent molecular pathways, with nodes representing pathway components and edges capturing their interactions. The continuous node states, encompassing variables like gene expression levels and protein concentrations, are interconnected through a system of ODEs, enabling the modeling of complex regulatory dynamics. ODE models are often available in the Systems Biology Markup Language (SBML), a standardized format that contains essential information about variables, parameters, equations, and model metadata in XML files.

To perform numerical simulations of ODE SBML models, we rely on the SBMLtoODEjax python library, a recent development that automates the parsing and conversion of SBML models into Python models written entirely in JAX (*Scheiner, 2019*). Taking advantage of JAX computing capabilities, SBMLtoODEjax enables efficient and parallel numerical solutions for gene expression levels and other node states by recursively invoking the generated Python models to integrate the ODE equations with current gene expression levels. Additionally, we have developed a Python library (copy archived at *Etcheverry et al., 2023a*) comprising additional modules and pipelines that facilitate interventions on the GRN models such as genome or drug interventions, as well as other perturbations such as noise, pushes, and walls that can be applied to the states and kinematic parameters of gene regulatory networks.

Given the model species initial state $y(t = 0)$, the desired rollout length $T(secs)$ and step size $\Delta T$, as well as the chosen intervention $i$ and/or perturbation $u$, the model rollout iteratively (1) integrates the system of ODE-governed equations that specifies the rate of species changes $\frac{dy}{dt}$ using JAX odeint solver to update model species $y(t) \rightarrow y(t + \Delta T)$, (2) calls the model assignment rules to update kinematic parameters if needed, and (3) apply the intervention and/or perturbation function to update $(y(t + \Delta T), w(t + \Delta T), c)$ accordingly. In this paper, we use $T = 2500s$ and $\Delta T = 0.1$ (25,001 time points per rollout including $t_0$). The ODE solver uses an absolute tolerance of $1e^{-6}$ and relative tolerance of $1e^{-12}$, with maximum number of solver steps of 1000. For a step-by-step guide on utilizing these libraries within the proposed framework, we refer interested readers to our tutorial (https://developmentalsystems.org/curious-exploration-of-grn-competencies/tuto1.html), which offers practical examples and detailed instructions.

### Experimental setup

In our computational models, we are able to record the activities of all nodes during a model rollout. The observation space $O \subset R_{+n \times \frac{T}{\Delta T}}$ is such that $o = (y(0), \ldots, y(T))$ where y(t) represents the n-dimensional vector of node states at each time step, with T being the total reaction time. The boundaries of the observation space are not known.

Regarding the exploration of problem spaces, namely the intervention space I and behavior space Z, we specify them as follows.

For the main experiments on biological networks, the intervention space $I \subset R_{+n}$ consists of initial node states sampled from the hyper-rectangle $[y_{0,min}, y_{0,max}]$ where $y_{0,min} = \frac{1}{r} \times y_{d,min}$ and $y_{0,max} = r \times y_{d,max}$ with $r = 20$ and $(y_{d,min}, y_{d,max})$ the minimum and maximum of each node of the model over the default time course simulation (with initial conditions provided in the SBML file and T=25000). On the other hand, the behavior space $Z \subset R_{+2}$ endpoint states $z = (y_i(T), y_j(T))$ where $(i,j)$ corresponds to the target phenotype nodes. We ensure that most trajectories have reached stable states at T=2500 (as elaborated in the next section) such that Z can be viewed as the space of reachable endpoints, whose boundaries are not known.

### Database creation

#### Biological networks database

All the ODE models we use in this work are downloaded from the BioModels database (*Glont, 2018*, *Malik-Sheriff et al., 2020*) in SBML format. From all models referenced on the website, we only

consider the ones that are curated, that have at least three nodes, and that are handled by the SBMLtoODEjax simulator (as SBMLtoODEjax does not handle models with discrete events, custom functions, or other specific cases as detailed in *Etcheverry et al., 2023b*). Note that the original models 37, 262, 263, 284, 454, 455, 459, 461, and 624 have been previously analyzed while clamping certain nodes at fixed values (as detailed in *Supplementary file 1*). Here, we relaxed this condition for a more realistic simulation in which all of the nodes' concentration are free to vary.

To ensure the inclusion of models suitable for our analyses, we then applied specific filters to the collected models.

First, we simulated the default model rollout for each model to obtain the concentration profiles of the pathway components over a short time span (T=10 s and $\Delta T = 0.1$). We discarded simulation results containing invalid values (NaN or negative concentrations) or those that took an excessive amount of time (>1 s). While it is acceptable that a rollout sometimes returns NaN values (when there are no solutions given ODE tolerance options for specific initial conditions), we consider the model invalid if this occurs for the default initial conditions provided in the SBML file.

For the remaining models, we conducted further simulations with an extended time span (T=2500) and 50 random initial conditions uniformly sampled within the model's intervention space $I$ (as defined before). Once again we discarded models whose batch simulations took an excessive amount of time (>15 s). From the remaining models, we derived the resulting 50 trajectories for each node pair (i, j) and subjected them to additional filters to refine the database. We removed node pairs where either (1) [filter F1] a substantial proportion of trajectories (20%) exhibited invalid concentrations (NaN or negative) or unsettled behaviors ($\exists t \geq 2400$ such that $|y(t) - y(T)| \geq 0.02 \times |y(T) - y(0)|$) or periodic patterns $\left( \exists f > 0 \, such \, that \, |S(f)| \geq 40 \, where \, S = DFT\left( \left[ y\left(\frac{T}{2}\right), \cdots, y(T) \right] \right) \right)$; or [filter F2] the reached space in $Z$ was too small $\left( \left( \max_{k=1 \cdots 50} y^k(T) - \min_{k=1 \cdots 5} y^k(T) \right) < 0.1 \right)$ to discard cases where 'diversity' could result from floating point rounding errors; or [filter F3] the number of attractors was less than four $\left( \left\{ y^k(T) \right\}_{k=1 \cdots 50} cover \leq 4 \frac{bins}{a} 20 \times 20 \, binning \, of \, Z \right)$.

Upon completion of the filtering process, our final database comprised 30 models, consisting of a total of 432 systems, as detailed in *Supplementary file 1*. These curated models and systems served as the foundation for our subsequent analyses and investigations into the navigation competencies of the molecular pathways.

## Random networks database

Following the methodology proposed in *Biswas et al., 2023*, we aimed to create a database of synthetic networks with topologies similar to those of the biological networks, but with random regulatory rules instead of evolved ones. The objective was to compare the versatility and robustness competencies between biological and random networks, akin to the approach used for memory competencies in *Biswas et al., 2023*. To achieve this, we initially generated 300 networks based on the transcriptional gene circuit model (*Reinitz and Sharp, 1995*), ensuring that they had the same distribution of network size (number of nodes) and connectivity (nodes in-degree) as the biological network database (using fitted Gaussian distributions). The kinematic parameters $W, b, \tau$ of these networks were randomized $\left( W \sim [-30, 30]^{n \times n}, B \sim [-10, 10]^n, \tau \sim [1, 15] \right)$ where model step is defined as $y(t+1) = \frac{\Delta T}{\tau} \times sigmoid(Wy + B) + \left( 1 - \frac{\Delta T}{\tau} \right) \times y$ and in-degree connectivity is enforced by setting some weights of $W$ to zero. However, during the creation process, we observed that none of the generated networks met the criterion for exhibiting a sufficient number of steady states (criterion F3). This limitation arose from the inherent constraints imposed by the gene circuit model's shape of ODE equations, limiting the diversity of possible dynamical behaviors. As our focus was on networks with a possible spectrum of steady states, akin to the biological network database, we decided not to pursue further analyses on these networks.

Instead, we selected the systems (models and pairs of nodes) that demonstrated the highest versatility (metric detailed below) from among all the generated systems that passed the filters F1 and F2. The selected networks' versatility is presented in *Figure 7*, but for future research, it would be interesting to explore broader and more complex classes of equations to assess their potential for achieving higher behavioral diversity.

## Sanity check

We also tested our exploration pipeline on two simple models, namely BIOMD0000000341 as described in the paper 'A model of beta-cell mass, insulin, and glucose kinetics: pathways to diabetes' (*Topp et al., 2000*) and BIOMD0000000454 as described in the 'Example One' of the 'Metabolic Control Analysis: Rereading Reder' paper (*Smallbone, 2013*). For both these models, the ground truth goal states (attractors of the models) are already known, because easy to find analytically or numerically as described in the corresponding papers. As a sanity check, we validate that our exploration pipeline is able to re-discover those goal states and added those results in *Figure 3—figure supplement 1* for completeness. Note that a more complex variant of BIOMD0000000454 is also included in our main biological networks database where we let the metabolite concentrations y1(t),... y5(t) evolve in time. We also illustrate results of our curiosity-driven exploration method and of a random search method on this more complex variant in the figure.

## Curiosity-driven exploration

This section provides additional information about the internal models and hyperparameters of the intrinsically-motivated goal exploration process. The overall IMGEP pipeline is illustrated in *Figure 1c*. To sample a goal, the IMGEP uses a uniform sampling strategy within the bounding hyper-rectangle of currently reached goals (scaled by a factor 1.3). Hence sampling bounds adapt to the discoveries and do not need to be predefined via expert knowledge. The volume of the hyper-rectangle is larger compared to the cloud of currently-reached goals, which incentivizes targeting unexplored areas outside of the cloud and promotes diversity in the exploration process. Then, to generate an intervention for achieving the sampled goal, the IMGEP selects the nearest previously reached goal in $Z$, identifies its associated intervention, and performs a local random step from that point $stepsize \sim N\left(0, 0.1 * \left[y_{0,max} - y_{0,min}\right]\right)$ in the intervention space.

While our implementation choices for the IMGEP goal representation, goal generation, and goal-conditioned optimization are relatively straightforward, it is worth noting that alternative strategies could be considered for each of these components for more complex problems. The Python library AutoDiscJax (copy archived at *Etcheverry et al., 2023a*) that accompanies this paper can be used to implement this and other IMGEP variants in JAX.

## Robustness tests

We define three family of perturbations: (1) the noise perturbation $U_n\left(\sigma_n, p_n \vee y\right)$ which is parametrized by its standard-deviation (scaled proportionally to the extent of the observed trajectory $y$ prior perturbation) and period (secs); (2) the push perturbation $U_p\left(m_p, n_p \vee y\right)$ parametrized by its magnitude (proportional to the extent of $y$) and number of occurrences; (3) the wall perturbation $U_w\left(l_w, n_w \vee y\right)$ parametrized by its length (proportional to the extent of $y$) and number, and where walls are generated in locations of the space that the GRN would 'naturally' visit without the induced perturbation. Details about the implementation of walls are provided in *Figure 5—figure supplement 1*.

To assess the robustness of the GRN systems in our database, we employ an evaluation procedure, as depicted in *Figure 1d*. For each system $(I, Z)$ in the database with its corresponding behavioral catalog $H$ discovered using the curiosity-search algorithm, we perform the following steps. We first retrieve $K$ representative trajectories out of the $N$ discoveries, i.e., ones that cover well the reachable space. To do so, we randomly sample tuples of K discoveries (among N) 500 times, and select the one with the maximum diversity. One could test all trajectories with K=N but here we use K=N/10 mainly for compute reasons, as we run the experimental campaign on all 432 systems. Next, we subject each of these K trajectories $\left\{y_k, k = 1..K\right\}$ to s=18 different perturbation distributions, each representing various levels of difficulty:$(\sigma_n, p_n) \in \left\{(0.001, 5), (0.005, 5), (0.1, 5), (0.005, 10), (0.005, 5), (0.005, 1)\right\}$, $(m_p, n_p) \in \left\{(0.05, 1), (0.1, 1), (0.15, 1), (1, 0.1), (2, 0.1), (3, 0.1)\right\}$, $(l_w, n_w) \in \left\{(0.05, 1), (0.1, 1), (0.15, 1), (1, 0.1), (2, 0.1), (3, 0.1)\right\}$. In each perturbation distribution, we sample r=3 random perturbations, resulting in $P = s * r$ perturbations. For each perturbation in the set $\left\{u_p, p = 1...P\right\}$, we re-run the trajectory starting from the same initial state $i$ but with the sampled perturbation applied $(i, u_p)$, and observe the resulting outcome $(o_p)$ and reached endpoint $(z_p)$.

At the end of this process, the behavioral catalog is augmented with the perturbed trajectories $H = \left\{\left(i_k, o_k, z_k, \left\{(u_p, o_p, z_p), p = 1...P\right\}\right), k = 1...K\right\}$.

## Evaluation metrics

### Diversity measure

Diversity is measured by the area that explored observations cover in behavior space Z. Each single exploration results in a new point in this space, such that diversity measures how much area the algorithms explored in those spaces.

In general, existing approaches in the NS, QD, and IMGEP literature use binning-based metrics (*Reinke et al., 2020*; *Etcheverry et al., 2020*; *Pugh et al., 2015*) or distance-based metric from ecology (*Scheiner, 2019*) to quantify the diversity of a set of explored instances. However, those metrics are sensitive to the binning strategy, or fail to discriminate between qualitatively significantly different explorations (*Benureau, 2015*). Another approach, called the threshold coverage, measures diversity as the volume of the union of the set of hyperballs of radius $\epsilon$ that have for centers the observed effects $\{z \in Z\}$. This diversity measure, while difficult to compute in high-dimensional spaces, avoids the pitfalls of bin-based and distance-based metrics and is easily computable in 2-dimensional spaces (*Benureau, 2015*).

Threshold coverage quantifies the area of the space that has been reached at a given precision $\epsilon$ (the threshold), and is what we used in *Figure 3* to compare random search and curiosity-driven exploration strategies.

### Sensitivity measure

In general, existing approaches in systems biology and evolutionary genetics measure sensitivity (opposite of robustness) in a relative manner with respect to (1) a functionality (*Kitano, 2007b*) or phenotypic trait (*Félix and Barkoulas, 2015*) of interest, (2) specific perturbations (environmental or genetic changes), and (3) a measure of the degree of variation. Here, we adopt a similar metric where (1) the phenotypic trait of interest is defined as a goal state $z \in Z$ discovered by curiosity search, (2) the set of perturbation $\{u_p\}$ is defined in previous section and conditioned on the GRN goal-reaching trajectory $i \rightarrow z$, and (3) variation is measured as the Euclidean distance in behavior space, normalized by the extent of the trajectory prior perturbation in Z.

This distance-based sensitivity measure proves straightforward as we explicitly use 'spaces' to observe and analyze behaviors. The results of this sensitivity analysis are presented in *Figure 5*.

### Versatility-robustness measure

In this study, we introduce the terms 'diversity' and 'versatility' to characterize the competencies of the exploration agent (IMGEP) and the GRN agent, respectively. Diversity refers to the ability of the IMGEP agent to reveal a wide range of behaviors in the GRN, while versatility refers to the capability of the GRN agent to reach diverse goal states. The GRN versatility is unknown, and can only be approximated via proxy metric. Here, we consider that the diversity of the IMGEP (measured with the threshold coverage metric) is a good approximation of the versatility of a given GRN, as the IMGEP was shown to efficiently drive the GRN into diverse possible goal states. In *Figure 7a*, we employ this diversity metric to categorize the versatility of surveyed networks based on the class of organism they belong to. For the random networks, as they all have less or equal than 4 attractors, the versatility remains below $0.026 = 4 \times \frac{\pi\epsilon^2}{(1+2\epsilon)^2}$.

*Figure 7b*, we introduce the versatility-robustness metric, which conditions the diversity metric on a sensitivity threshold. Only goal states with sensitivity to perturbations below this threshold are considered when computing the reached area of the space. A high versatility-robustness score indicates that diverse goal states are achieved with a high level of precision.

## Experiments on the RKIP-ERK signaling pathway

This section details the additional experiments conducted on the RKIP-ERK signaling pathway (*Lehman and Stanley, 2011*). We refer to the accompanying notebook tutorial for reproducing these experiments: https://developmentalsystems.org/curious-exploration-of-grn-competencies/tuto1.html.

For *Figure 4*, clustering in behavior space was performed using the HDBSCAN algorithm (*McInnes et al., 2017*) with hyperparameters set as min_cluster_size = 10 and cluster_selection_epsilon = 0.1. Points in the 10-dimensional intervention space are visualized by applying a TSNE 2-dimensional reduction. To visualize the clusters in behavior space (and corresponding clusters in intervention

space), we fitted polygons on the cluster points using shapely library unary_union, dilatation, and erosion operations (*Pugh et al., 2015*; *Gillies, 2022*).

In *Figure 6*, we generated trajectory-based energy landscapes following the method proposed in *Venkatachalapathy et al., 2021*. Energy landscapes provide an intuitive way to understand how a system with multiple steady states behave, by picturing it as a ball rolling downhill towards low-energy valleys (steady states). Given a set of trajectories in behavior space Z, we constructed a probability distribution (P) of system states and converted it into a pseudopotential energy surface (U = −ln(P)). This energy surface was smoothed using cubic spline interpolation and visualized using Plotly 3D surface plots. *Figure 6a, b, c* differed by the input set of trajectories used for generating the landscape: (a) employed the set of trajectories discovered by random search, (b) used the set of trajectories discovered by curiosity search, and (c) utilized the set of trajectories generated by robustness tests.

In *Figure 8*, the 'healthy' and 'disease' clusters were the same as in *Figure 4* and visualized similarly. We displayed trajectories with the lowest sensitivity (averaged over all $P = 3 \times 18$ perturbations). The stimuli-based intervention shown in *Figure 8b* was found using a simple random search procedure. First, we defined an arbitrary target node and a stepwise node-activation function, clamping MKEPP values to desired values $x = \left[ y_{MEKPP}^{(1)}, \cdots, y_{MEKPP}^{(10)} \right]$ every 10 s for 100 s. Then, we randomly sampled x within a range of values near the MKEPP current steady states (endpoints from the 6 'disease' trajectories, assuming that the drug intervention cannot drastically remodel those values). For each candidate x, we ran new trajectories starting from the disease states and applying the intervention x under a distribution of noise, push, and wall perturbations. Finally, we selected the intervention x that most successfully brought ERK-RKIP levels back to the target setpoint (centroid of the healthy region). The resulting intervention (shown in *Figure 8b*) succeeds to robustly reset all 6 disease state points despite perturbations, as shown in *Figure 8c*. We refer to the notebook for reproducing the experiments.

## Experiments on synthetic gene networks

This section details the additional experiments conducted on the synthetic gene networks (*Figure 9*). We refer to the second accompanying tutorial for the full codebase: https://developmentalsystems. org/curious-exploration-of-grn-competencies/tuto2.html.

In these experiments, we consider the target application of gene circuit engineering followed in *Hiscock, 2019*, where parameters of a gene circuit model are optimized to produce target oscillator patterns. The gene circuit model employed in *Hiscock, 2019* is the same than the one used for the random networks database (Eq 1), with $\tau = 1$. Hence the *intervention space* is a $n^2 + 2n$ dimensional space defined as $I = \left[ y_{t=0,min}, y_{t=0,max} \right] \oplus \left[ W_{min}, W_{max} \right] \oplus \left[ B_{min}, B_{max} \right]$, with $y_{0,min} = 0, y_{0,max} = 1, W_{min} = -30, W_{max} = 30, B_{min} = -10, B_{max} = 10$. Here, we consider networks of n=3 nodes, with the first node being the target phenotype node. Thus, what we seek here is kinematic parameters $(W, B)$ and initial concentrations $y_0$ that would produce a periodic pattern $y = \left[ y_{n=0}(0), \cdots, y_{n=0}(T) \right]$ with target amplitude $A$, frequency $w$ and offset $b$. Here, the target $(A, \omega, b)$ are sample randomly with $A \sim U\left( [0.1, 0.5] \right), b \sim U\left( [A, 1 - A] \right), \omega \sim Beta\left( \alpha = 2, \beta = 8 \right)$.

We then compare three alternative exploration strategies: (1) curiosity search, (2) random search and (3) gradient descent, i.e., pure optimization-driven search as proposed in *Hiscock, 2019*, all given the same experimental budget $N = 5000$.

For curiosity search, the behavior space $Z$ is defined as the image space of the discrete Fourier transform of the 1d-signal $y$, where distance in the space measures average difference in spectral amplitude. The IMGEP algorithm is then the same that the one previously used, as detailed in *Figure 1c*, but operating within the novel problem spaces $(I, Z)$.

For random search, interventions are sample uniformly $(i_1, \cdots, i_N) \sim U(I)$.

For gradient descent, we follow the procedure proposed in *Hiscock, 2019*. We define a loss function which, for a set of parameters $i = (W, B, y_0)$, measures the mean square error between the phenotype node activation levels $y$ and the target oscillation represented as a cosine wave with the desired $(A, \omega, b) : L = \sum_t \left( y(t) - \left( A\cos(2\pi\omega t) + b \right) \right)^2$. We then sample a random parameter $i \sim U(I)$ and use

Adam optimizer with $l_r = 10^{-3}, b1 = 0.02, b3 = 0.001, \epsilon = 10^{-8}$ for $N = 5000$ optimization steps (same number of model rollouts allowed than for curiosity search and random search).

In addition, we use gradient descent for *local* refinement of the best discoveries made by the other exploration strategies (curiosity search and random search), this time with a limited budget of $N = 100$ optimization steps.

Visualizations in *Figure 9* show: (a-b) the oscillators discovered by random search and curiosity search (gradient descent did not find any oscillator in this example) in the $(A, \omega, b)$ space, (c) the corresponding diversity (using this time a binning-based space coverage measure with $20^3$ bins as the space is 3-dimensional), (d) the evolution of the training loss $L$ throughout the n=5000 trials for the three exploration strategies, (e-f-g) the corresponding best discoveries (for which $L$ is minimal) for the three exploration strategies, and (h-i) the local training loss and resulting finetuning of the best discoveries with gradient descent.

## Statistics

Statistical analyses were performed using custom code in Python 3.9 (relevant libraries include jax 0.4.8 and scipy 1.10 and plotly 5.16.1). Welch's two-sample t-tests reported in *Figure 3* are two-tailed, and additional details on statistical analyses are provided in the figure legend.

## Acknowledgements

We thank Patrick Erickson and Randall Jordan Ellis for review and discussion, as well as Tom Cirrito, Wesley Clawson, and Santosh Manicka for useful discussions. We also thank Alexander Mordvinstev for providing the executable paper template, as well as Julia Poirier for assistance with the manuscript. The authors acknowledge support from the biotechnology company Poietis and the French National Association of Research and Technology (ANRT), as well as from the French National Research Agency (ANR, DeepCuriosity AI chair project). ML gratefully acknowledges funding support from Astonishing Labs, and from the Templeton World Charity Foundation via grant TWCF0606. This work also benefited from the use of the Jean Zay supercomputer associated with the Genci grant A0151011996.

## Additional information

### Competing interests

Michael Levin: lab receives funding in the form of a sponsored research agreement from, and consults for, Astonishing Labs, which has interest in therapeutic applications of GRN learning behaviors. The other authors declare that no competing interests exist.

### Funding

| Funder | Grant reference number | Author |
|---|---|---|
| Poietis Company | | Mayalen Etcheverry |
| French National Association of Research and Technology | | Mayalen Etcheverry |
| French National Research Agency | DeepCuriosity AI chair project | Pierre-Yves Oudeyer |
| Astonishing Labs | | Michael Levin |
| Templeton World Charity Foundation | TWCF0606 | Michael Levin |
| Genci | A0151011996 | Mayalen Etcheverry Clément Moulin-Frier Pierre-Yves Oudeyer |

| Funder | Grant reference number | Author |
|--------|------------------------|--------|

The funders had no role in study design, data collection and interpretation, or the decision to submit the work for publication.

## Author contributions

Mayalen Etcheverry, Conceptualization, Data curation, Software, Formal analysis, Funding acquisition, Validation, Investigation, Visualization, Methodology, Writing – original draft, Writing – review and editing; Clément Moulin-Frier, Pierre-Yves Oudeyer, Michael Levin, Conceptualization, Resources, Supervision, Funding acquisition, Methodology, Writing – original draft, Project administration, Writing – review and editing

## Author ORCIDs

Mayalen Etcheverry ⓘ http://orcid.org/0000-0001-9568-6081
Michael Levin ⓘ https://orcid.org/0000-0001-7292-8084

Reviewer #1 (Public Review): https://doi.org/10.7554/eLife.92683.4.sa1
Reviewer #2 (Public Review): https://doi.org/10.7554/eLife.92683.4.sa2
Author response https://doi.org/10.7554/eLife.92683.4.sa3

# Additional files

## Supplementary files

MDAR checklist

Supplementary file 1. List of biological networks from Biomodels used in this paper.

## Data availability

Source code is available on GitHub (copy archived at *Etcheverry et al., 2024*). It contains experimental data and an executable notebook version of the paper to reproduce all paper figures. It also contains additional step-by-step tutorials to reproduce results from scratch for Figures 4, 6 and 8 and Figure 9, as well as the codebase to reproduce the whole experimental campaign. All our codebase is open-source under MIT License. The ODE models we use in this work are downloaded from the BioModels database, and ID references of these models are detailed in *Supplementary file 1*.

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
