## [Editor Report · eLife assessment]

This **important** study develops a machine learning method to reveal hidden unknown functions and behaviors in gene regulatory networks by searching parameter space in an efficient way. **Solid** evidence is presented for the method, which should be of broad interest to anyone working in biology, as the ideas put forward by the authors extend beyond gene regulatory networks to reveal hidden functions in any complex system with many interacting parts.

---

## [Referee Report · Reviewer #1 (Public Review)]

Summary:

This paper suggests to apply intrinsically-motivated exploration for the discovery of robust goal states in gene regulatory networks.

Strengths:

The paper is well written. The biological motivation and the need for such methods are formulated extraordinarily well. The battery of experimental models is impressive.

Weaknesses:

(1) The proposed method is compared to the random search. That says little about the performance with regard to the true steady-state goal sets. The latter could be calculated at least for a few simple ODE (e.g., BIOMD0000000454, `Metabolic Control Analysis: Rereading Reder'). The experiment with 'oscillator circuits' may not be directly interpolated to the other models.

The lack of comparison to the ground truth goal set (attractors of ODE) from arbitrary initial conditions makes it hard to evaluate the true performance/contribution of the method. A part of the used models can be analyzed numerically using JAX, while there are models that can be analyzed analytically.

"...The true versatility of the GRN is unknown and can only be inferred through empirical exploration and proxy metrics....": one could perform a sensitivity analysis of the ODEs, identifying stable equilibria. That could provide a proxy for the ground truth 'versatility'.

(2) The proposed method is based on `Intrinsically Motivated Goal Exploration Processes with Automatic Curriculum Learning', which assumes state action trajectories [s_{t_0:t}, a_{t_0:t}], (2.1 Notations and Assumptions' in the IMGEP paper). However, the models used in the current work do not include external control actions, but rather only the initial conditions can be set. It is not clear from the methods whether IMGEP was adapted to this setting, and how the exploration policy was designed w/o actual time-dependent actions. What does "...generates candidate intervention parameters to achieve the current goal...."

mean considering that interventions 'Sets the initial state...' as explained in Table 2?

(3) Fig 2 shows the phase space for (ERK, RKIPP_RP) without mentioning the typical full scale of ERK, RKIPP_RP. It is unclear whether the path from (0, 0) to (~0.575, ~3.75) at t=1000 is significant on the typical scale of this phase space. is it significant on the typical scale of this phase space?

(4) Table 2:

(a) Where is 'effective intervention' used in the method?

(b) In my opinion 'controllability', 'trainability', and 'versatility' are different terms. If there correspondence is important I would suggest to extend/enhance the column "Proposed Isomorphism". otherwise, it may be confusing. I don't see how this table generalizes generalizes "concepts from dynamical complex systems and behavioral sciences under a common navigation task perspective".

---

## [Referee Report · Reviewer #2 (Public Review)]

Summary:

Etcheverry et al. present two computational frameworks for exploring the functional capabilities of gene regulatory networks (GRNs). The first is a framework based on intrinsically motivated exploration, here used to reveal the set of steady states achievable by a given gene regulatory network as a function of initial conditions. The second is a behaviorist framework, here used to assess the robustness of steady states to dynamical perturbations experienced along typical trajectories to those steady states. In Figs. 1-5, the authors convincingly show how these frameworks can explore and quantify the diversity of behaviors that can be displayed by GRNs. In Figs. 6-9, the authors present applications of their framework to the analysis and control of GRNs, but the support presented for their case studies is often incomplete.

Following revision, my overall perspective of the paper remains unchanged. The first half of the paper provides solid evidence to support an important conceptual framework. The evidence presented for the use cases in the latter half is incomplete; as the authors note, they are preliminary and meant to be built on in future work. I have included my first round comments below.

Strengths:

Overall, the paper presents an important development for exploring and understanding GRNs/dynamical systems broadly, with solid evidence supporting the first half of their paper in a narratively clear way.

The behaviorist point of view for robustness is potentially of interest to a broad community, and to my knowledge introduces novel considerations for defining robustness in the GRN context.

Some specific weaknesses, mostly concerning incomplete analyses in the second half of the paper:

(1) The analysis presented in Fig. 6 is exciting but preliminary. Are there other appropriate methods for constructing energy landscapes from dynamical trajectories in gene regulatory networks? How do the results in this particular case study compare to other GRNs studied in the paper?

Additionally, it is unclear whether the analysis presented in Fig. 6C is appropriate. In particular, if the pseudopotential landscapes are constructed from statistics of visited states along trajectories to the steady state, then the trajectories derived from dynamical perturbations do not only reflect the underlying pseudo-landscape of the GRN. Instead, they also include contributions from the perturbations themselves.

(2) In Fig. 7, I'm not sure how much is possible to take away from the results as given here, as they depend sensitively on the cohort of 432 (GRN, Z) pairs used. The comparison against random networks is well-motivated. However, as the authors note, comparison between organismal categories is more difficult due to low sample size; for instance, the "plant" and "slime mold" categories each only has 1 associated GRN. Additionally, the "n/a" category is difficult to interpret.

(3) In Fig. 8, it is unclear whether the behavioral catalog generated is important to the intervention design problem of moving a system in one attractor basin to another. The authors note that evolutionary searches or SGD could also be used to solve the problem. Is the analysis somehow enabled by the behavioral catalog in a way that is complementary to those methods? If not, comparison against those methods (or others e.g. optimal control) would strengthen the paper.

(4) The analysis presented in Fig. 9 also is preliminary. The authors note that there exist many algorithms for choosing/identifying the parameter values of a dynamical system that give rise to a desired time series. It would be a stronger result to compare their approach to more sophisticated methods, as opposed to random search and SGD. Other options from the recent literature include Bayesian techniques, sparse nonlinear regression techniques (e.g. SINDy), and evolutionary searches. The authors note that some methods require fine-tuning in order to be successful, but even so, it would be good to know the degree of fine-tuning which is necessary compared to their method. [second round: the authors have included a comparison against CMA-ES, an evolutionary algorithm]

---

## [Author Response]

The following is the authors’ response to the previous reviews.

**Recommendations for the authors:**

**Reviewer #1 (Recommendations For The Authors):**
My main concern is still in place. It is unclear whether the proposed method can find actual goal states, and as a result it is unclear what states it finds. Table S1 mentions the model BIOMD0000000454, which is a small metabolic pathway with known equations given in "Example One" in "Metabolic Control Analysis: Rereading Reder". In this model the goal states can be calculated analytically.Regarding your statements below: I am not concerned that your method will be less efficient than random search (or any other search..) on small models, but I think it is important for the readers to have evidence that your method is able to discover true goal states at least in small networks, used in your study. You do show that your method scales to complex models. So, in my opinion, the missing part is to show that it is able to find true goal states."...For simple models whose true steady-state distribution can be derived numerically and/or analytically, it is very likely that their exploration will be much simpler and this is not where a lot of improvement over random search may be found, which explains our focus on more complex models..."

We thank you for your response and for your concerns on the lack of evidence that our method is able to re-discover the true goal states of simple models when these are known a priori. We acknowledge that adding these simple cases is useful for completeness. We did not include these simple models in our main study because in most cases a basic random search over the initial conditions will lead to the re-discovery of these goal states. For instance for the mentioned model BIOMD0000000454 described in the "Example One" from the "Metabolic Control Analysis: Rereading Reder" paper, several simplifying assumptions are made such that the system only has one steady state (x1=0.056, x2=0.769, x3=4.231) which can be found analytically as shown in the paper. In that simple case, this goal state is also straightforward to find with numerical simulation as any valid initial condition will converge to it.

To address the concerns of the reviewer, we propose to add an additional "sanity check" figure in the supplementary of the revised paper (Figure S4), as well as a “sanity check” subsection in the “Methods”, to present additional experiments made on simple models such as this one. The novel figure and subsection can be visualized on the paper’s interactive version available online https://developmentalsystems.org/curious-exploration-of-grn-competencies, and we plan to include them as such in the further revision. We have also included the full code to reproduce this sanity check as a ‘sanity_check.ipynb’ jupyter notebook in the github repository (https://github.com/flowersteam/curious-exploration-of-grn-competencies/blob/main/notebooks/sanity_check.ipynb).

In the novel figure S4-b, we show the results of our exploration pipeline on the suggested model BIOMD0000000454 as described in the "Example One" of the paper. These results provide evidence that the curiosity search is able to find back the correct unique goal state (x1=0.056, x2=0.769, x3=4.231), as expected.

We also include a second sanity check on BIOMD0000000341 which models the dynamics of beta-cell mass, insulin and glucose dynamics. This model has two stable fixed points representing physiological (B=300, I=10, G=100) and pathological (B=0, I=0, G=600) steady states, which are the known ground truth steady states as described in Figure 3 of the "A Model of b-Cell Mass, Insulin, and Glucose Kinetics: Pathways to Diabetes" paper. Again, as expected, curiosity search is able to find back those two steady states (Figure S4-a).

As stated in our previous answer, our main study focuses on more complex models that are not limited to one or few attractors that can easily be discovered with random initial conditions. Regarding the mentioned BIOMD0000000454, maybe something that has been confusing for the reviewer is that we indeed included it in our main study but, as specified in the caption of table S4, at the difference of what is done in the "example one" of the original paper, we let the metabolite concentrations y1,...,y5 evolve in time (instead of enforcing them as constants). When doing so, the resulting dynamics of the system are more complex and exhibit a spectrum of possible steady states (unknown a priori), which differ from the previous case with a single steady state. In that case, the new attractors are not analytically easy to find and the proposed curiosity search becomes interesting as it is able to uncover the distribution of possible steady states much more efficiently than a random search baseline, as shown in the new figures S4-c and S4-d.

We hope that these new results will address the reviewer’s concerns and provide evidence to the readers on the validity of the approach on simple networks.

**eLife assessment**
This important study develops a machine learning method to reveal hidden unknown functions and behavior in gene regulatory networks by searching parameter space in an efficient way. The evidence for some parts of the paper is still incomplete and needs systematic comparison to other methods and to the ground truth, but the work will be of broad interest to anyone working in biology of all stripes since the ideas reach beyond gene regulatory networks to revealing hidden functions in any complex system with many interacting parts.

We thank the editors and reviewers for their positive assessment and constructive suggestions. In our response, we acknowledge the importance of systematic comparison to other methods and to the ground truth, when available. However we also emphasize the challenges associated with evaluating such methods in the context of uncovering hidden behaviors in complex biological networks as the ground truth is often unknown. We hope that our explanations will clarify the potential of our approach in advancing the exploration of these systems.

**Public Reviews:**

**Reviewer #1 (Public Review):**
Summary: This paper suggests to apply intrinsically-motivated exploration for the discovery of robust goal states in gene regulatory networks.Strengths:The paper is well written. The biological motivation and the need for such methods are formulated extraordinarily well. The battery of experimental models is impressive.

We thank the reviewer for sharing interest in the research problem and for recognizing the strengths of our work.

Weaknesses:(1) The proposed method is compared to the random search. That says little about the performance with regard to the true steady-state goal sets. The latter could be calculated at least for a few simple ODE (e.g., BIOMD0000000454, `Metabolic Control Analysis: Rereading Reder'). The experiment with 'oscillator circuits' may not be directly interpolated to the other models.The lack of comparison to the ground truth goal set (attractors of ODE) from arbitrary initial conditions makes it hard to evaluate the true performance/contribution of the method. A part of the used models can be analyzed numerically using JAX, while there are models that can be analyzed analytically."...The true versatility of the GRN is unknown and can only be inferred through empirical exploration and proxy metrics....": one could perform a sensitivity analysis of the ODEs, identifying stable equilibria. That could provide a proxy for the ground truth 'versatility'.

We agree with the reviewer that one primary concern is to properly evaluate the effectiveness of the proposed method. However, as we move toward complex pathways, knowledge of the “true” steady-state goal sets is often unknown which is where the use of machine learning methods as the one we propose are particularly interesting (but challenging to evaluate).

For simple models whose true steady-state distribution can be derived numerically and/or analytically, it is very likely that their exploration will be much simpler and this is not where a lot of improvement over random search may be found, which explains our focus on more complex models. While we agree that it is still interesting to evaluate exploration methods on these simple models for checking their behavior, it is not clear how to scale this analysis to the targeted more complex systems.

For systems whose true steady state distribution cannot be derived analytically or numerically, we believe that random search is a pertinent baseline as it is commonly used in the literature to discover the attractors/trajectories of a biological network. For instance, Venkatachalapathy et al. [1] initialize stochastic simulations at multiple randomly sampled starting conditions (which is called a kinetic Monte Carlo-based method) to capture the steady states of a biological system. Similarly, Donzé et al. [29] use a Monte Carlo approach to compute the reachable set of a biological network «when the number of parameters is large and their uncertain range is not negligible». For the considered models, the true steady-state goal set is unknown, which is why we chose comparison with random search. We added a “Statistics” subsection in the Methods section providing additional details about the statistical analyses we perform between our method and the random search baseline.

(2) The proposed method is based on `Intrinsically Motivated Goal Exploration Processes with Automatic Curriculum Learning', which assumes state action trajectories [s_{t_0:t}, a_{t_0:t}], (2.1 Notations and Assumptions' in the IMGEP paper). However, the models used in the current work do not include external control actions, but rather only the initial conditions can be set. It is not clear from the methods whether IMGEP was adapted to this setting, and how the exploration policy was designed w/o actual time-dependent actions. What does "...generates candidate intervention parameters to achieve the current goal....", mean considering that interventions 'Sets the initial state...' as explained in Table 2?

We thank the reviewer for asking for clarification, as indeed the IMGEP methodology originates from developmental robotics scenarios which generally focus on the problem of robotic sequential decision-making, therefore assuming state action trajectories as presented in Forestier et al. [65]. However, in both cases, note that the IMGEP is responsible for sampling parameters which then govern the exploration of the dynamical system. In Forestier et al. [65], the IMGEP also only sets one vector at the start (denoted) which was specifying parameters of a movement (like the initial state of the GRN), which was then actually produced with dynamic motion primitives which are dynamical system equations similar to GRN ODEs, so the two systems are mathematically equivalent. More generally, while in our case the “intervention” of the IMGEP (denoted) only controls the initial state of the GRN, future work could consider more advanced sequential interventions simply by setting parameters of an action policy at the start which could be called during the GRN’s trajectory to sample control actions where would be the state of the GRN. In practice this would also require setting only one vector at the start, so it would remain the same exploration algorithm and only the space of parameters would change, which illustrates the generality of the approach.

(3) Fig 2 shows the phase space for (ERK, RKIPP_RP) without mentioning the typical full scale of ERK, RKIPP_RP. It is unclear whether the path from (0, 0) to (~0.575, ~3.75) at t=1000 is significant on the typical scale of this phase space. is it significant on the typical scale of this phase space?

The purpose of Figure 2 is to illustrate an example of GRN trajectory in transcriptional space, and to illustrate what “interventions” and “perturbations” can be in that context. To that end we have used the fixed initial conditions provided in the BIOMD0000000647, replicating Figure 5 of Cho et al. [56].

While we are not sure of what the reviewer means with “typical” scale of this phase space, we would like to point reviewer toward Figure 8 which shows examples of certain paths that indeed reach further point in the same phase space (up to ~10 in RKIPP_RP levels and ~300 in ERK levels). However, while the paths displayed in Figure 8 are possible (and were discovered with the IMGEP), note that they may be “rarer” to occur naturally in the sense that a large portion of the tested initial conditions with random search tend to converge toward smaller (ERK, RKIPP_RP) steady-state values similar to the ones displayed in Figure 2.

(4) Table 2:a. Where is 'effective intervention' used in the method?b. in my opinion 'controllability', 'trainability', and 'versatility' are different terms. If their correspondence is important I would suggest to extend/enhance the column "Proposed Isomorphism". otherwise, it may be confusing.

a) We thank the reviewer for pointing out that “effective intervention” is not explicitly used in the method. The idea here is that as we are exploring a complex dynamical system (here the GRN), some of the sampled interventions will be particularly effective at revealing novel unseen outcomes whereas others will fail to produce a qualitative change to the distribution of discovered outcomes. What we show in this paper, for instance in Figure 3a and Figure 4, is that the IMGEP method is particularly sample-efficient in finding those “effective interventions”, at least more than a random exploration. However we agree that the term “effective intervention” is ambiguous (does not say effective in what) and we have replaced it with “salient intervention” in the revised version.

b) We thank the reviewer for highlighting some confusing terms in our chosen vocabulary, and we have clarified those terms in the revised version. We agree that controllability/trainability and versatility are not exactly equivalent concepts, as controllability/trainability typically refers to the amount to which a system is externally controllable/trainable whereas versatility typically refers to the inherent adaptability or diversity of behaviors that a system can exhibit in response to inputs or conditions. However, they are both measuring the extent of states that can be reached by the system under a distribution of stimuli/conditions, whether natural conditions or engineered ones, which is why we believe that their correspondence is relevant.

I don't see how this table generalizes "concepts from dynamical complex systems and behavioral sciences under a common navigation task perspective".

We have replaced the verb “generalize” with “investigate” in the revised version.

**Reviewer #2 (Public Review):**
Summary:Etcheverry et al. present two computational frameworks for exploring the functional capabilities of gene regulatory networks (GRNs). The first is a framework based on intrinsically-motivated exploration, here used to reveal the set of steady states achievable by a given gene regulatory network as a function of initial conditions. The second is a behaviorist framework, here used to assess the robustness of steady states to dynamical perturbations experienced along typical trajectories to those steady states. In Figs. 1-5, the authors convincingly show how these frameworks can explore and quantify the diversity of behaviors that can be displayed by GRNs. In Figs. 6-9, the authors present applications of their framework to the analysis and control of GRNs, but the support presented for their case studies is often incomplete.Strengths:Overall, the paper presents an important development for exploring and understanding GRNs/dynamical systems broadly, with solid evidence supporting the first half of their paper in a narratively clear way.The behaviorist point of view for robustness is potentially of interest to a broad community, and to my knowledge introduces novel considerations for defining robustness in the GRN context.

We thank the reviewer for recognizing the strengths and novelty of the proposed experimental framework for exploring and understanding GRNs, and complex dynamical systems more generally. We agree that the results presented in the section “Possible Reuses of the Behavioral Catalog and Framework” (Fig 6-9) can be seen as incomplete along certain aspects, which we tried to make as explicit as possible throughout the paper, and why we explicitly state that these are “preliminary experiments”. Despite the discussed limitations, we believe that these experiments are still very useful to illustrate the variety of potential use-cases in which the community could benefit from such computational methods and experimental framework, and build on for future work.

Some specific weaknesses, mostly concerning incomplete analyses in the second half of the paper:(1) The analysis presented in Fig. 6 is exciting but preliminary. Are there other appropriate methods for constructing energy landscapes from dynamical trajectories in gene regulatory networks? How do the results in this particular case study compare to other GRNs studied in the paper?

We are not aware of other methods than the one proposed by Venkatachalapathy et al. [1] for constructing an energy landscape given an input set of recorded dynamical trajectories, although it might indeed be the case. We want to emphasize that any of such methods would anyway depend on the input set of trajectories, and should therefore benefit from a set that is more representative of the diversity of behaviors that can be achieved by the GRN, which is why we believe the results presented in Figure 6 are interesting. As the IMGEP was able to find a higher diversity of reachable goal states (and corresponding trajectories) for many of the studied GRNs, we believe that similar effects should be observable when constructing the energy landscapes for these GRN models, with the discovery of additional or wider “valleys” of reachable steady states.

Additionally, it is unclear whether the analysis presented in Fig. 6C is appropriate. In particular, if the pseudopotential landscapes are constructed from statistics of visited states along trajectories to the steady state, then the trajectories derived from dynamical perturbations do not only reflect the underlying pseudo-landscape of the GRN. Instead, they also include contributions from the perturbations themselves.

We agree that the landscape displayed Fig. 6C integrates contributions from the perturbations on the GRN’s behavior, and that it can shape the landscape in various ways, for instance affecting the paths that are accessible, the shape/depth of certain valleys, etc. But we believe that qualitatively or quantitatively analyzing the effect of these perturbations on the landscape is precisely what is interesting here: it might help (1) understand how a system respond to a range of perturbations and to visualize which behaviors are robust to those perturbations, (2) design better strategies for manipulating those systems to produce certain behaviors

(2) In Fig. 7, I'm not sure how much is possible to take away from the results as given here, as they depend sensitively on the cohort of 432 (GRN, Z) pairs used. The comparison against random networks is well-motivated. However, as the authors note, comparison between organismal categories is more difficult due to low sample size; for instance, the "plant" and "slime mold" categories each only have 1 associated GRN. Additionally, the "n/a" category is difficult to interpret.

We acknowledge that this part is speculative as stated in the paper: “the surveyed database is relatively small with respect to the wealth of available models and biological pathways, so we can hardly claim that these results represent the true distribution of competencies across these organism categories”. However, when further data is available, the same methodology can be reused and we believe that the resulting statistical analyses could be very informative to compare organismal (or other) categories.

(3) In Fig. 8, it is unclear whether the behavioral catalog generated is important to the intervention design problem of moving a system from one attractor basin to another. The authors note that evolutionary searches or SGD could also be used to solve the problem. Is the analysis somehow enabled by the behavioral catalog in a way that is complementary to those methods? If not, comparison against those methods (or others e.g. optimal control) would strengthen the paper.

We thank the reviewer for asking to clarify this point, which might not be clearly explained in the paper. Here the behavioral catalog is indeed used in a complementary way to the optimization method, by identifying a representative set of reachable attractors which are then used to define the optimization problem. For instance here, thanks to the catalog, we (1) were able to identify a “disease” region and several possible reachable states in that region and (2) use several of these states as starting points of our optimization problem, where we want to find a single intervention that can successfully and robustly reset all those points, as illustrated in Figure 8. Please note that given this problem formulation, a simple random search was used as an optimization strategy. When we mention more advanced techniques such as EA or SGD, it is to say that they might be more efficient optimizers than random search. However, we agree that in many cases optimizing directly will not work if starting from random or bad initial guess, and this even with EA or SGD. In that case the discovered behavioral catalog can be useful to better initialize this local search and make it more efficient/useful, akin to what is done in Figure 9.

(4) The analysis presented in Fig. 9 also is preliminary. The authors note that there exist many algorithms for choosing/identifying the parameter values of a dynamical system that give rise to a desired time-series. It would be a stronger result to compare their approach to more sophisticated methods, as opposed to random search and SGD. Other options from the recent literature include Bayesian techniques, sparse nonlinear regression techniques (e.g. SINDy), and evolutionary searches. The authors note that some methods require fine-tuning in order to be successful, but even so, it would be good to know the degree of fine-tuning which is necessary compared to their method.

We agree that the analysis presented in Figure 9 is preliminary, and thank the reviewer for the suggestion. We would first like to refer to other papers from the ML literature that have more thoroughly analyzed this issue, such as Colas et al. [74] and Pugh et al. [34], and shown the interest of diversity-driven strategies as promising alternatives. Additionally, as suggested by the reviewer, we added an additional comparison to the CMA-ES algorithm in the revised version in order to complete our analysis. CMA-ES is an evolutionary algorithm which is self-adaptive in the optimization steps and that is known to be better suited than SGD to escape local minimas when the number of parameters is not too high (here we only have 15 parameters). However, our results showed that while CMA-ES explores more the solution space at the beginning of optimization than SGD does, it also ultimately converges into a local minima similarly to SGD. The best solution converges toward a constant signal (of the target b) but fails to maintain the target oscillations, similar to the solutions discovered by gradient descent. We tried this for a few hyperparameters (init mean and std) but always found similar results. We have updated the figure 9 image and caption, as well as descriptive text, to include these novel results in the revised version. We also added a reference to the CMA-ES paper in the citations.

**Reviewer #1 (Recommendations For The Authors):**
I would suggest to conduct a more rigor analysis of the performance by estimating/approximating the ground truth robust goal sets in important GRNs.Also, the use of terminology from different disciplines can be improved. Please see my comments above. Specifically, the connection between controllability in dynamical control systems and versatility used in this paper is unclear.

We hope to have addressed the reviewer's concerns in our previous answers.

**Reviewer #2 (Recommendations For The Authors):**
Fig 4b: I'm not sure if DBSCAN is the appropriate method to use here, as the visual focus on the core elements of the clusters downplays the full convex hull of the points that random sampling achieves in Z space. An analysis based on convex hulls or the ball-coverage from Fig. 3b would presumably generate plots that were more similar between random sampling and curiosity search. If the goal is to highlight redundancy/non-linearity in the mapping between Z and I, another approach might be to simply bin Z-space in a grid, or to use a clustering algorithm that is less stringent about core/noise distinctions.

We thank the reviewer for the suggestion. This plot is intended to convey the reader an understanding of why a method that uniformly samples goals in Z (what the IMGEP is doing), is more efficient than a method that uniformly samples parameters in I (what the random search is doing), in systems for which there is high redundancy/non-linearity in the mapping between I and Z. We agree that binning the Z-space in a grid and counting the number of achieved bins is a way to quantitatively measure this, which is by the way very close to what we do in Figure 3 for measuring the achieved diversity. We believe however that the clustering and coloring provides additional intuitions on why this is the case: it illustrates that large regions of the intervention space map to small regions in the outcome space and vice versa.

Additional changes in the revised version:

We added a sentence in the Methods section as well as in the caption of Table S1 providing additional details about the way we simulate the biological models from the BioModels website

We fixed a wrong reference to Figure 4 in the Methods “Sensitivity measure” subsection with reference to Figure 5.